



# Creation and analysis of a multi-hazard database: Tenerife (Canary Islands) as a case study

Marta López-Saavedra[1], Joan Martí[1], Marc Martínez-Sepúlveda[1]

[1]Natural Risks Assessment and Management Service (NRAMS), Institute of Environmental Assessment and Water Research
(IDAEA-CSIC), Barcelona, 08034, Spain

*Correspondence to*: Marta López-Saavedra (geomartalopez@gmail.com)

**Abstract.** In the context of escalating climate change impacts and the heightened frequency of natural hazards, the imperative for robust multi-risk assessment and proactive mitigation strategies has become increasingly evident. Tenerife, situated in the Atlantic archipelago, encapsulates the challenges faced by communities globally, prompting a paradigm shift
towards anticipatory risk management. This study presents a pioneering effort to establish a multi-hazard database for regions susceptible to be impacted by multiple natural hazards, using Tenerife (Canary Islands) as a case study, to provide a foundation for more accurate risk assessments and informed decision-making. Our methodology involved the systematic collection and analysis of over 500 years of historical data (https://doi.org/10.20350/digitalCSIC/17088) on volcanic activity, earthquakes, floods, landslides, and extreme weather events, allowing us to identify patterns, vulnerabilities, and effective
resilience measures. In this sense, our holistic approach aims to empower stakeholders with a nuanced understanding of natural processes. The database reveals key patterns in hazard occurrence and impacts, such as the frequent and damaging floods linked to heavy rainfall and ravine overflows. It also highlights the increasing frequency and severity of these events in recent decades, underscoring the urgent need for improved management practices. Other hazards, like rock falls and landslides, though less common, pose significant risks in areas affected by human activities. Key recommendations include
the implementation of flood prevention measures such as ravine cleaning, water retention areas, and reforestation, as well as enhanced geotechnical studies and slope stabilization efforts to mitigate landslide risks. The approach outlined here is not only applicable to Tenerife but serves as a scalable model for other regions facing complex natural hazard scenarios. By leveraging historical insights alongside contemporary methodologies, this contribution aims to strengthen natural risk resilience and inform future risk mitigation strategies.

**Keywords:** multi-hazard, database, long-term hazard assessment, multi-risk management, Tenerife, natural hazards, innovative approach, natural risk resilience, mitigation.

## 1 Introduction

In the face of escalating climate change impacts and the increasing frequency and intensity of natural hazards, the need for robust multi-hazard risk assessment and mitigation strategies has never been more critical. Fortunately, contemporary





discourse on risk management has evolved beyond a reactive stance, prompting a paradigm shift toward proactive and anticipatory approaches (e.g., UNISDR, 2015; UNDRR, 2022). Many regions worldwide have a complex history of exposure to various natural hazards, including volcanic activity, seismic events, and meteorological phenomena (e.g., López-Saavedra & Martí, 2023; Lee et al., 2024). Understanding the dynamic interplay of these hazards is fundamental to predicting and mitigating their future manifestations (e.g., Ming et al., 2022; Ward et al., 2022; López-Saavedra & Martí,

35  2023).

Traditionally, risk assessments have relied on physics-based predictive models. However, data-driven predictive models are increasingly being integrated into risk assessments (Stødle et al., 2023). Data-driven modeling is instrumental in approximating the relationships between explanatory and response variables (Stødle et al., 2023). This empirical approach not only enables the identification of complex patterns but also provides a framework for addressing uncertainties—an

essential component of multi-hazard assessments, given the intricate and often unexpected interrelationships among natural phenomena (López-Saavedra & Martí, 2023). Today, the continuous collection of vast amounts of operational data through advanced sensing and monitoring systems allows for more sophisticated, data-driven risk assessment approaches (Guikema, 2020). These methodologies, which include statistical and machine learning models, leverage extensive datasets to predict future outcomes and evaluate potential risks, thereby surpassing the limitations imposed by historical data alone (Stødle et

al., 2023).

Despite the advancements facilitated by data-driven methods, a significant challenge remains: the availability and comprehensiveness of historical and real-time data on natural phenomena in specific regions. Many areas face limitations in both the instrumentation used to monitor these phenomena and the relatively recent initiation of data collection efforts (UNDRR & WMO, 2023), which restricts the availability of long-term datasets necessary for robust risk assessment and

management. In numerous regions, existing datasets are insufficiently comprehensive to establish baseline conditions or accurately define "background" levels of natural hazards. This lack of historical data undermines the ability to set effective warning thresholds and develop comprehensive risk management protocols. For instance, without a detailed historical record, identifying patterns or establishing reliable baselines for event frequency and intensity becomes challenging, thereby hindering the development of accurate risk models and effective hazard preparedness strategies (Stødle et al., 2023).

Addressing these data gaps necessitates the development of a comprehensive historical record of all significant events that have occurred in each region. The significance of such a database extends beyond data collection, as it provides a critical resource for developing and validating predictive models applicable across different scenarios and timeframes. For example, probabilistic risk models, which simulate a range of possible future events based on historical data and expert knowledge, depend on the quality and completeness of the available data (UNDRR, 2012). Additionally, probabilistic modelling can be

employed to generate deterministic scenarios by using real data as input values, which are then subjected to climate change predictive models to assess hazard impacts. A well-documented multi-hazard database supports the generation of robust probabilistic risk assessments and enhances the capacity to manage uncertainties associated with future risk scenarios (UNDRR, 2012). Such a record serves as a foundation for understanding the full spectrum of potential hazards and their



impacts, thereby enabling more informed and proactive risk management strategies (Guikema, 2020). Furthermore, a
comprehensive historical dataset not only facilitates the establishment of baseline conditions but also supports the
development of early warning systems and response protocols by providing a detailed account of past events, which is
instrumental in evaluating future risks (López-Saavedra & Martí, 2023). Therefore, establishing a thorough historical record
of natural hazards is indispensable for effective multi-hazard assessments and for enhancing the resilience of communities
against future events. By compiling a comprehensive dataset, regions can better identify historical patterns, develop accurate
risk models, and design more effective risk management strategies, in alignment with the principles outlined in the Sendai
Framework for Disaster Risk Reduction (UNISDR, 2015).

This approach aims not only to document a region's encounters with various hazards but also to empower stakeholders with
a comprehensive understanding of the underlying natural processes. This holistic strategy, integrating contemporary
monitoring techniques with historical context, establishes a synergy essential for effective, long-term risk mitigation (e.g.,
Notti et al., 2022; Luino, Barriendos et al., 2023; Luino, Gizzi et al., 2023; Oliva & Olcina, 2024). This contribution serves
as a blueprint for developing a multi-hazard database that integrates data from diverse sources and in various formats.
Tenerife has been selected as a case study due to its status as a microcosm of vulnerability within the Atlantic archipelago,
exemplifying the challenges faced by communities worldwide. This approach systematically captures the historical and
geological record of natural events. The proposed methodology not only illuminates the unique challenges faced by this
region but also provides a scalable model applicable to areas experiencing diverse natural phenomena. By establishing this
comprehensive database, we aim to facilitate precise multi-hazard assessments and improve multi-risk management. Our
ultimate objective is to contribute to a proactive and informed approach, enhancing natural risk resilience in Tenerife and
serving as a template for regions worldwide that grapple with the complex interplay of natural hazards affecting their
populations.

## 2 Geological setting and natural hazards

With a land area of 2,034 km², Tenerife is the largest of the eight volcanic islands that constitute the Canary Islands
archipelago. Located in the east-central Atlantic Ocean, approximately 300 km off the southern coast of Morocco in
northwest Africa (Fig. 1a), the Canary Islands lie within the African Plate. This archipelago, associated with a persistent
mantle plume, has been the subject of ongoing debate regarding its structure and geodynamic evolution (Hernández-Pacheco
& Ibarrola, 1973; Anguita & Hernán, 1975, 2000; Schmincke, 1982; Araña & Ortiz, 1991; Hoernle & Schmincke, 1993;
Carracedo et al., 1998; Fullea et al., 2015).

Regional tectonics in this sector of the African Plate, investigated through GPS measurements, seismicity analysis, and
crustal deformation studies (Jiménez-Munt et al., 2001; Jiménez-Munt & Negredo, 2003; Serpelloni et al., 2007; Jiménez-
Munt et al., 2011; Cunha et al., 2012; Bezzeghoud et al., 2014), have revealed the presence of a simple extensional
deformation field oriented west to east across the Canary Archipelago. This deformation, which is perpendicular to the Mid-

Atlantic and Terceira Ridges, has been identified as a key mechanism facilitating the formation of pathways for magma ascent to the surface (Anguita & Hernán, 2000).



**Figure 1: (a) Geographical location of the Canary Islands in general, and Tenerife in particular, together with the distribution of municipalities and urban areas; (b) schematic cross-section of the island. The black lines correspond to the cross-section of part (b) of this figure. Source: adapted from López-Saavedra et al. (2021) and López-Saavedra et al. (2023), (CC BY-NC-ND 4.0).**

Centrally located within the archipelago, Tenerife emerges as a large, pyramid-shaped volcanic edifice, rising nearly 8,000 m

above the Miocene oceanic crust (Fig. 1b). The island hosts the active Teide volcano, which reaches an elevation of 3,718 m above sea level, ranking Tenerife as the third-largest and one of the most complex volcanic systems globally, following Mauna Loa and Mauna Kea in Hawaii. Despite its geographical distance from the Iberian Peninsula, the Canary Islands remain part of the Spanish state due to their colonial history.

The geological evolution of Tenerife is characterized by two primary superimposed volcanic complexes: a basaltic shield

complex (>12 Ma to present; Abdel-Monem et al., 1972; Ancochea et al., 1990; Thirlwall et al., 2000) and a central complex (<4 Ma to present; Fuster et al., 1968; Araña, 1971; Ancochea et al., 1990; Martí et al., 1994) (Fig. 1b). As an active volcanic island, Tenerife is subject to volcanic hazards. The construction of the island has been primarily driven by basaltic shield-building volcanism, which remains active today through eruptions along the northeastern (Dorsal Rift Zone) and northwestern (Santiago Rift Zone) fissure vent systems. Additionally, volcanic activity occurs at several small scoria-lava

cones located at the heads of major landslide valleys (La Orotava, Icod, and Güímar) and within a broad monogenetic volcanic field in the southern part of the island (Martí, 2019).

Historical eruptions on Tenerife include those of Siete Fuentes (1704), Fasnia (1705), Arafo (1706), Garachico (1706), Chahorra (1798), and Chinyero (1909) (Fig. 2). In contrast, Teide and its twin volcano, Pico Viejo, have not erupted in historical times, with their last known eruption (Lavas Negras) occurring approximately 1,000 years ago (Martí et al., 2022).

However, up to 18 eruptions have been identified on the island during the Holocene (Martí et al., 2008).

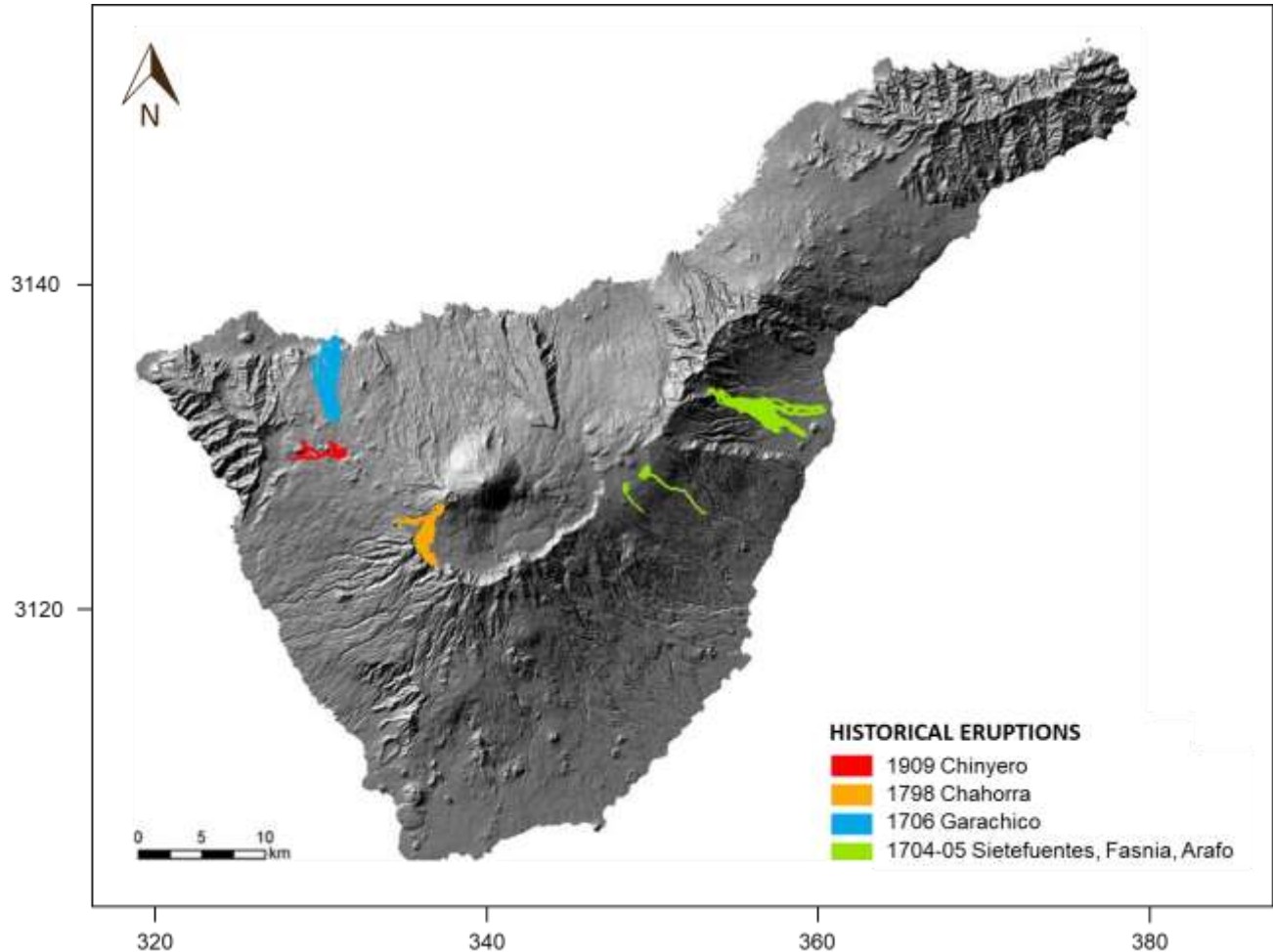

**Figure 2. Map of the historical eruptions of Tenerife (Canary Islands). Source: adapted from IGN (n.d.).**

Seismic activity in Tenerife is closely associated with volcanic processes, as evidenced by the high-magnitude earthquakes that occurred during the triple eruption of Siete Fuentes, Fasnia, and Arafo (1704–1705) (Sánchez-Sanz, 2014). In fact, all historical eruptions have been preceded and/or accompanied by felt earthquakes (Romero, 1991).

At present, moderate-magnitude (typically not exceeding 2.0 mbLg(L), occasionally reaching 3.0 mbLg(L)) and shallow seismicity, primarily related to magma intrusion and hydrothermal system dynamics, is concentrated along the Santiago Rift Zone, particularly around the Chahorra (1798) and Chinyero (1909) volcanoes. Similar activity is also observed along the Dorsal Rift Zone, affecting the Arafo, Fasnia, and Siete Fuentes (1704–1705) volcanoes, as well as the southern monogenetic volcanic field (IGN, 2023a,b).

In addition to this background seismicity, Tenerife experiences episodic seismic swarms that are concentrated in four key seismogenic zones: (i) the western sector of the Caldera de Las Cañadas, where two significant swarms, comprising hundreds of low-magnitude earthquakes, occurred in 2016 and 2019; (ii) the Izaña region, which exhibited substantial

seismic activity between 2009 and 2011; (iii) the area surrounding Pico del Teide, where seismicity recurs periodically; and

(iv) the Vilaflor region, at the head of the southern monogenetic volcanic field, where intermittent seismic activity has persisted to the present (Domínguez Cerdeña et al., 2019).

However, the most significant earthquakes have been concentrated along the fault zone between Tenerife and Gran Canaria (Fig. 3), which has produced the strongest seismic event recorded in historical times (ML 5.3 in 1989; Mezcua et al., 1992).

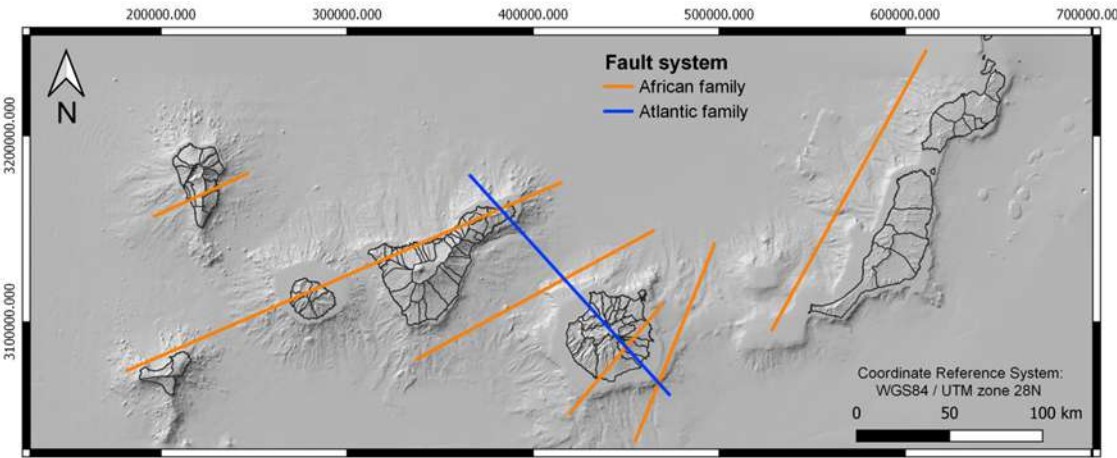

**Figure 3. Volcano-tectonic lines of the Canary Islands' region. Source: adapted from Bosshard & Macfarlane (1970), and Mezcua et al. (1992). *Note: Orange lines = African family of faults; blue lines = Atlantic family of faults. Fault traces are not to scale nor is the location accurate, but are indicative.**

Beyond these endogenous geological processes, Tenerife has been affected by multiple additional hazards that have resulted in economic losses and, on occasion, human casualties. These include landslides of varying magnitudes, floods, forest fires,

and even tsunamis.

## 3 Natural hazards, risk management and risk regulations

The Territorial Insular Emergency Plan for Civil Protection of the Island of Tenerife (PEIN) (Cabildo de Tenerife, 2020) identifies several natural hazards, including hydrological risks, seismic activity, volcanic eruptions, adverse atmospheric phenomena, slope movements, locust plagues, and forest fires. Additionally, the PEIN conducts a comparative risk analysis

to classify these hazards based on their probability of occurrence and potential consequences, thereby establishing priorities for planning and response strategies. This classification is based on historical data regarding hazardous events that have impacted the island's territory (hazard assessment) and an evaluation of vulnerable elements (vulnerability assessment), including the population, critical infrastructure, cultural heritage, and protected areas.

Risk assessment considers two key parameters: probability (the estimated or projected frequency of an event) and severity

(the extent of potential damage). The risk level is quantitatively determined using an index (1) that integrates these factors, expressed as follows:



$$Risk\ Index\ (RI) = Probability\ Index\ (PI) \times Severity\ Index\ (SI) \qquad (1)$$

The PI and SI can be as follows (Table 1):

**Table 1. Risk Probability and Severity Indices. Source: Cabildo de Tenerife (2020).**

| Probability of occurrence of the hazard | | Severity of risk | |
|---|---|---|---|
| 0 | Practically zero | 0 | No damage |
| 2 | Very low. No constancy | 1 | Minor material damage |
| 3 | Low. One occurrence every several years. | 2 | Minor material damage and/or some people affected |
| 4 | Medium. Every few years (less than 10 years) | 5 | Large material damage and/or many people affected |
| 5 | High. Once or several times per year | 10 | Major material damage and/or fatalities |


Based on the value obtained by multiplying each PI and SI assigned to each hazard, risks can be categorized according to their RI as follows (Table 2):

**Table 2. Risk Index (RI). Source: Cabildo de Tenerife (2020).**

| LOW | $0 \geq IR \geq 5$ | Minimal or virtually no risk |
|---|---|---|
| MEDIUM | $6 \geq IR \geq 8$ | A risk to be considered in the PEIN |
| HIGH | $10 \geq IR \geq 15$ | It is recommended that specific civil protection measures be adopted within the PEIN |
| VERY HIGH | $20 \geq IR \geq 50$ | In addition to the recommendations included in the PEIN, reference is made to the Special Plan corresponding to the risk in question |

Thus, the following risk classification included in the PEIN has been obtained (Table 3):

**Table 3. Classification of the Level of Risk of Different Natural Phenomena in Tenerife According to Their Probability and Severity. Source: Cabildo de Tenerife (2020).**

| NATURAL RISKS | | | | | |
|---|---|---|---|---|---|
| **Type of risk** | **Phenomenon** | **PI** | **SI** | **RI** | **Level of risk** |
| Hydrological risks | Floods | 4 | 10 | 40 | Very high |
| | Ruptures of large storage infrastructures | 2 | 5 | 10 | High |
| Seismic movements | Earthquakes | 3 | 2 | 6 | Medium |
| | Tsunamis | 3 | 2 | 6 | Medium |
| Volcanic eruptions | | 3 | 10 | 30 | Very high |
| Adverse atmospheric phenomena | Snowfall | 5 | 1 | 5 | Low |
| | Torrential rains | 5 | 10 | 50 | Very high |
| | Hailstorms and frost | 5 | 1 | 5 | Low |
| | Strong winds | 5 | 5 | 25 | Very high |
| | Coastal storms | 5 | 2 | 10 | High |
| | Heat waves | 5 | 2 | 10 | High |
| | Haze and dust in suspension | 5 | 5 | 25 | Very high |
| | Droughts | 3 | 5 | 15 | High |
| Slope movements | Rockfall | 5 | 2 | 10 | High |
| | Landslides | 4 | 2 | 8 | Medium |
| | Coastal erosion | 4 | 2 | 8 | Medium |
| Locust pests | | 2 | 2 | 4 | Low |
| Forest fires | | 5 | 10 | 50 | Very high |



According to this classification (Table 2 and Table 3), forest fires and torrential rainfall represent the greatest risks to the population of Tenerife, given their high frequency and significant impact. These are followed by floods, volcanic eruptions,
strong winds, and Saharan dust storms (haze). In contrast, snowfall, hailstorms, frost, and locust plagues are identified as the least significant risks. However, this risk probability must be periodically reassessed in light of historical records and the potential recurrence of geological events observed in the past.

Beyond these identified natural hazards, Tenerife also faces a critical challenge due to its insular nature: sea level rise. Given its geographical setting, this phenomenon is of particular concern, as it threatens coastal urban areas and infrastructure.
NASA has developed sea level rise projections based on data from the IPCC Sixth Assessment Report (Fox-Kemper et al., 2021; Garner et al., 2021; IPCC, 2022; Garner et al., 2023). These projections, available through an online interactive tool (https://sealevel.nasa.gov/ipcc-ar6-sea-level-projection-tool), provide estimates relative to the 1995–2014 baseline for five Shared Socioeconomic Pathway (SSP) scenarios and five future Global Mean Surface Temperature (GMST) levels (2080–2100). Sea level projections are also provided at five specific future Global Mean Surface Temperatures (from 2080-2100):
1.5°C, 2°C, 3°C, 4°C and 5°C.

According to these projections, in the worst-case scenario (SSP5-8.5, Fig. 3), Tenerife could experience a sea level rise of 1.53 m, whereas in the best-case scenario (SSP1-1.9, Fig. 3), the rise would be 0.75 m. This represents a serious threat, as most urban centers are located along the island's coastline (Fig. 1a). Furthermore, Tenerife is undergoing rapid population growth, necessitating further urban expansion, which could lead to increased exposure to other natural hazards in inland
areas.

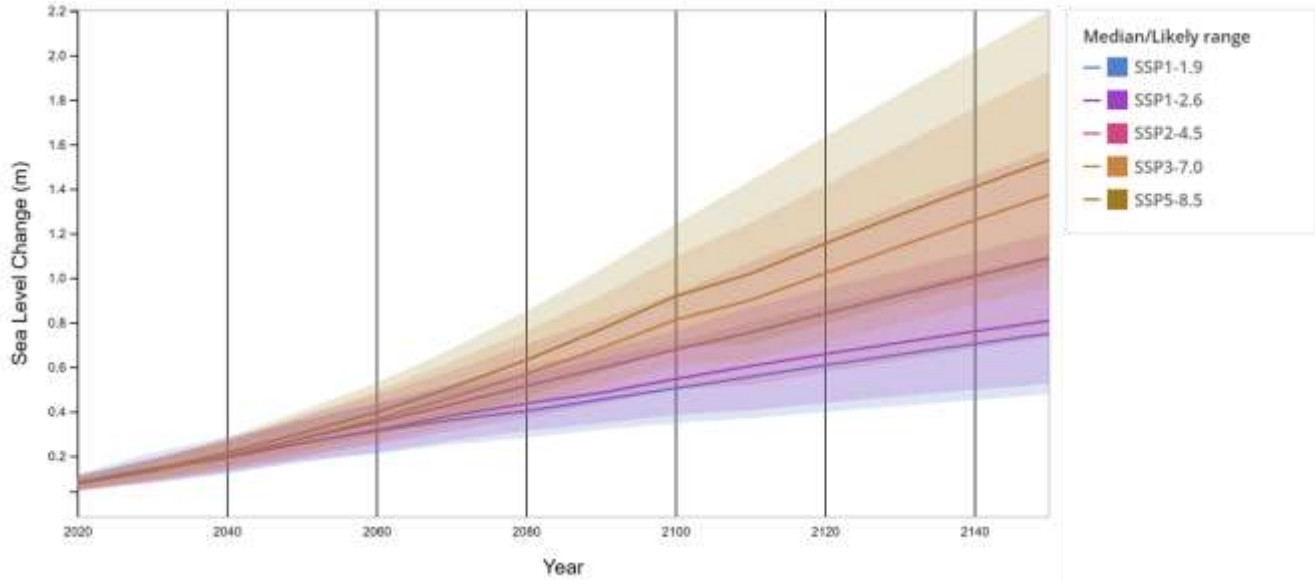

**Figure 4. Projected sea level rise under different SSP scenarios for Tenerife. Source: Fox-Kemper et al. (2021), Garner et al. (2021), Garner et al. (2023), Creative Commons Attribution 4.0 International License.**





*Note: By 2100, the SSP1-1.9 scenario limits global warming to roughly 1.5°C above pre-industrial levels (1850–1900),
following a temporary overshoot, and anticipates achieving net-zero $CO_2$ emissions around mid-century. The SSP1-2.6
pathway maintains warming below 2.0°C (median estimate), with carbon neutrality projected in the latter half of the
century. SSP2-4.5 aligns with the higher range of emission levels expected from current Nationally Determined
Contributions (NDCs) by 2030. The IPCC Special Report on 1.5°C (SR1.5) estimated that NDC implementation could lead
to a temperature increase between 2.7°C and 3.4°C by 2100, which falls within the upper bounds of SSP2-4.5 projections.
Although several countries introduced 2050 net-zero targets by the end of 2020—consistent with SSP1-1.9 and SSP1-2.6—
the overall emissions outlook for 2030 remained largely unchanged. The SSP2-4.5 trajectory modestly diverges from a
baseline scenario with no additional climate policies, with a median projected warming of approximately 2.7°C by century's
end. SSP3-7.0 represents a medium-to-high emissions pathway lacking new climate interventions, marked by substantial
non-$CO_2$ emissions such as aerosols. SSP5-8.5 constitutes a high-end reference scenario also assuming no further climate
policies, and such emission levels are only realized in Integrated Assessment Models under the fossil-fueled development
storyline of SSP5. Uncertainty ranges (17th–83rd percentiles) are included, and temperature projections are referenced to a
1995–2014 baseline. The accompanying figure illustrates projected total sea level change and associated uncertainties.
Source: Fox-Kemper et al. (2021), Garner et al. (2021), Garner et al. (2023).*

In addition to sea level rise, climate change is expected to alter Tenerife's climatic patterns, affecting average temperature
and precipitation levels. The Canary Islands exhibit a subtropical climate, characterized by mild winters (mean temperature
>20 °C), low annual precipitation (<225 mm), and abundant sunshine (2800 h/year) (Azorín-Molina et al., 2018; Megías &
García-Román, 2022). However, due to its location, the weather variability in the Canary Islands is influenced by the
interaction between the semi-permanent Azores subtropical high-pressure system, and its relation with the Icelandic Low,
and the air masses coming from the Sahara (Cropper & Hanna, 2014; Cabildo de Tenerife, 2020; Megías & García-Román,
2022). For this reason, there are three main types of weather on the islands (Cabildo de Tenerife, 2020):

1. The trade winds regime, characterized by stable, warm and not very rainy weather, which originates mainly in
   summer, when the Azores Anticyclone withdraws towards the Portuguese coast.
2. The Atlantic squalls, which leave more unstable and rainy weather, especially in autumn and spring, originating
   when the Azores Anticyclone withdraws towards the center of the Atlantic and a Polar Front squall approaches.
3. Saharan weather, warmer and drier, especially in the winter months, although it can occur at any time of the year,
   and is caused when the Azores Anticyclone withdraws towards the center of the Atlantic and a dry air mass from
   the Sahara arrives. It is often accompanied by the well-known "calima" or Saharan dust in suspension.

Tenerife's climate is strongly influenced by topography and orientation (Cabildo de Tenerife, 2020; Megías & García-
Román, 2022), resulting in pronounced contrasts between the windward (northern) and leeward (southern) slopes. The
northern slopes, exposed to the trade winds, experience higher precipitation, greater humidity, and less sunshine, whereas the
southern slopes are much drier and more arid. There are also great differences within each slope depending on the altitude



(e.g., precipitation and average annual temperature, respectively, of 223 mm and 21 ℃ on the coast of Santa Cruz de Tenerife, 559 mm and 16.8 ℃ in the middle zone of Los Rodeos, and 487 mm and 10 ℃ on the summits of Izaña) (Cabildo de Tenerife, 2020) (Fig. 4 and Fig. 5).

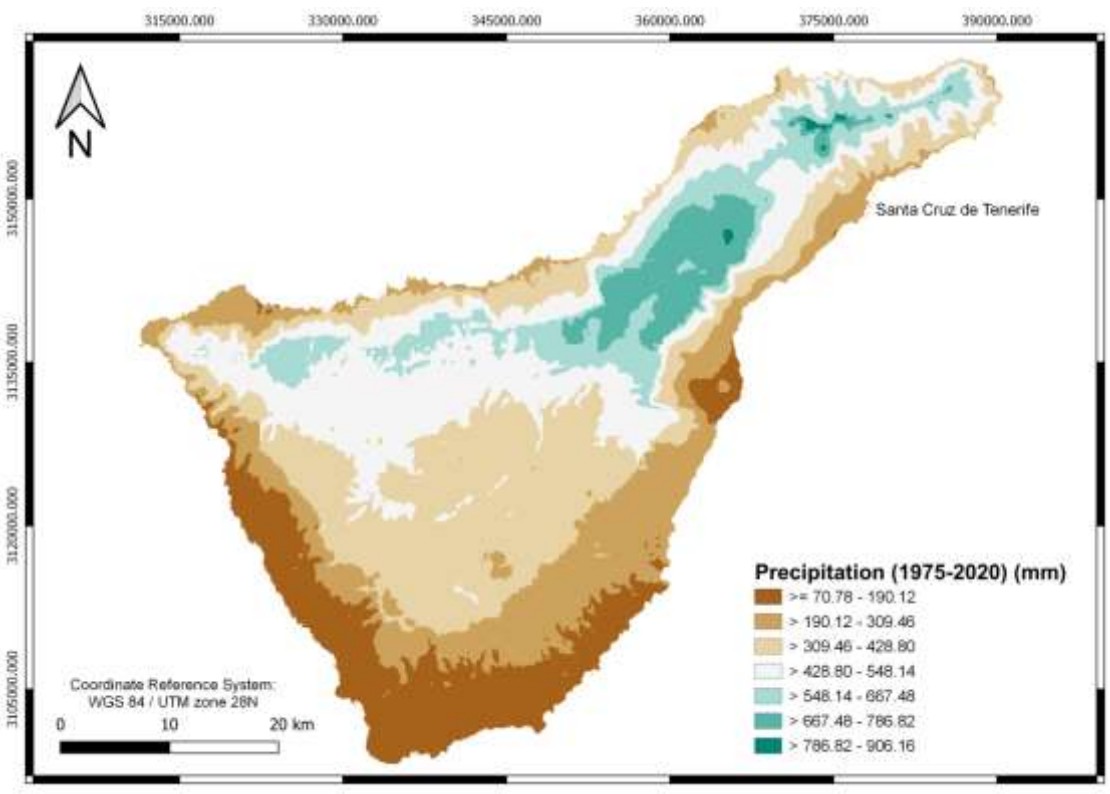


**Figure 5. Precipitation of Tenerife (Canary Islands) during the period 1975-2020 measured in millimeters (mm). Source: adapted from the Government of the Canary Islands (2023a).**



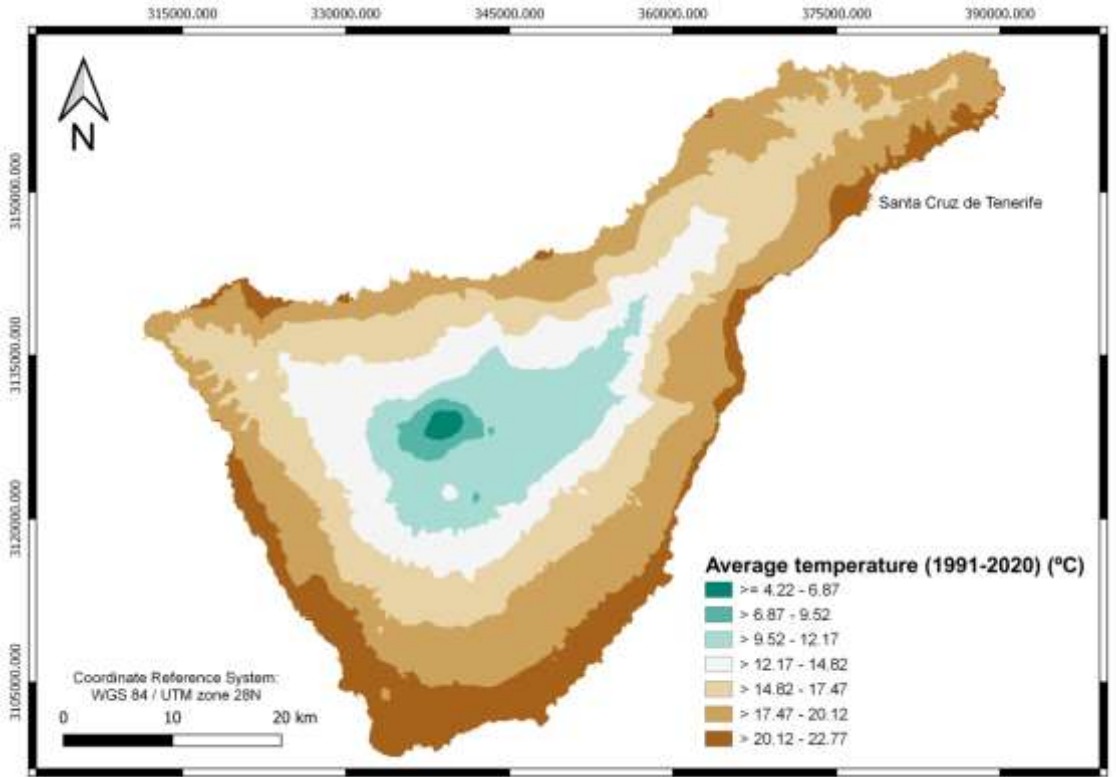

**Figure 6. Average temperature of Tenerife (Canary Islands) during the period 1991-2020 measured in degrees Celsius (ºC).**
**Source: adapted from Government of the Canary Islands (2023b).**

According to IPCC WGI Interactive Atlas data (https://interactive-atlas.ipcc.ch/) (Gutiérrez et al., 2021; Iturbide et al., 2021), under a 1.5°C global temperature increase, Tenerife's average temperature could rise by 1.1°C (SSP5-8.5), while annual precipitation could decrease by 18.5%. These projected changes may exacerbate existing natural hazards, including forest fires, torrential rainfall, flash floods, and haze, in addition to reducing water reserves.

While climate change does not directly affect the frequency or magnitude of volcanic eruptions or earthquakes, it may worsen associated secondary hazards. For instance, desertification and increased soil erosion could destabilize slopes, making them more prone to landslides during seismic events or volcanic unrest.

Given these risks, it is crucial to thoroughly analyze past natural hazards in Tenerife to develop effective risk mitigation strategies and enhance resilience in the face of future climate-related transformations.

**4 Methodology**

We compiled all existing records of historical events that have impacted the island of Tenerife, covering the period from its colonization by the Castilians around 1494 to the present day. This compilation was conducted as part of López-Saavedra's





doctoral thesis (2023) and was completed in 2022. Consequently, the dataset spans from 1494—established as the starting point for the reasons previously outlined—to 2020, which was selected as the endpoint in López-Saavedra's work (2023).

This time frame provides a sufficiently broad period to ensure high-quality results. The complete record is presented in Table A of the Supplementary Material.

For this study, we focused on geological hazards. Therefore, the event categories in our dataset included volcanic eruptions, earthquakes, landslides, floods, and tsunamis. For each event, we collected data on the following variables:

- Event (e.g., type, specific name, etc.)
- Start and end dates
- Location (i.e., place of origin)
- Cascading effects and associated hazards
- Main affected areas
- Fatalities
- Injuries
- Displacements
- Economic, social, and environmental losses
- Management and resilience measures (actions taken before, during, and after the event, as well as recovery efforts)
- Additional observations (e.g., measurement data, magnitudes, relevant notes)

As a documentary foundation, we relied on previously compiled historical records for each type of event. For volcanic eruptions, we used the Global Volcanism Program database from the Smithsonian Institution (2013) and studies by Carracedo (2008) and Romero (1991). For earthquakes, we consulted the Catalog and Seismic Bulletins for the Canary Islands, maintained by the National Geographic Institute (IGN, 2021), and supplemented this with the Review of the Seismic Catalog of the Canary Islands (1341–2000) (IGN, 2020). The dataset was limited to epicenters within latitudes 29° and 27°

and longitudes -15° and -18°, covering Tenerife and its offshore surroundings. Additionally, we filtered earthquakes by magnitude, selecting those with $mbLg(L) \geq 3.5$, as they are considered to release sufficient energy to be perceived by the population, potentially causing damage or serving as precursors to volcanic eruptions.

For landslides, we used the Movements Database of the Geological and Mining Institute of Spain (IGME, 2016). For floods, we consulted event catalogs from Arroyo (2009), Quirantes et al. (1993), Dorta (2007), and Pinto (1954), as well as the Plan

de Defensa Frente Avenidas de Tenerife (PDA) from the Tenerife Water Council (Consejo Insular de Aguas de Tenerife, 2004). Finally, for tsunamis, we referenced the study by Galindo et al. (2021).

Beyond these sources, we updated and expanded the records primarily through historical chronicles housed in the Provincial Historical Archive of Santa Cruz de Tenerife, the Diocesan Historical Archive of San Cristóbal de La Laguna, and the Municipal Historical Archive of San Cristóbal de La Laguna. Additionally, we examined documents from the Digital Press

Archive (Jable) of the University of Las Palmas de Gran Canaria (ULPGC) and the Virtual Library of Historical Press,

managed by the Subdirectorate General of Library Coordination of the Ministry of Culture and Sport of the Government of Spain. These documentary sources were further complemented by numerous scientific articles, historical newspaper archives, and official documents issued by the Cabildo of Tenerife and other islands. We also incorporated legal documents published in the Official State Gazette (BOE) of Spain, including Royal Decrees, Laws, and Orders enacted in response to

past hazards. The complete bibliography is listed as a Supplement.

Cross-referencing multiple sources enabled us to verify the dataset for inconsistencies, errors, or missing values, thereby improving its reliability and accuracy.

The historical record of Tenerife underwent a preliminary qualitative analysis. We evaluated the frequency and duration of each type of event, as well as their locations and areas of impact. We qualitatively identified recurring patterns or trends by

analyzing commonalities in the temporal and spatial distribution of events over the studied period. Additionally, we assessed the socioeconomic consequences of these events to determine their severity, considering economic losses, approved recovery budgets, the type and quantity of damaged infrastructure, and the extent of the damage (e.g., flooded, burnt, damaged, destroyed, devastated, affected). Finally, we analyzed the risk management measures implemented during and after these events, evaluating their effectiveness in mitigating impacts and enhancing resilience.

The historical record of multi-hazard events analyzed in this study was compiled into a database, which is publicly available in the DIGITAL.CSIC repository under the https://doi.org/10.20350/digitalCSIC/17088 (López-Saavedra, et al., 2025).

## 5 Results

From 1494 to 2020, a total of six volcanic eruptions occurred in Tenerife, two of which could be considered part of the same eruptive event, resulting in a total of five distinct occurrences. Additionally, 96 seismic events with a magnitude greater than

3.5 mbLg(L) were recorded during this period, 13 of which—whether seismic swarms or individual earthquakes—were considered of volcanic origin due to their association with previous eruptions (Fig. 6). Furthermore, 104 floods and five tsunamis were documented, most of which were triggered by distal earthquakes, except for one caused by a landslide on the coast of Tenerife.

Regarding landslides and rockfalls, the available records cover a much shorter period. Between 1941 and 2020, up to 198

such events were documented, although the exact date remains unknown for 71 of them. According to these data, landslides and rockfalls could be considered the most frequent geological hazards, followed by floods. However, it is important to note that only earthquakes with a magnitude greater than 3.5 mbLg(L) were considered due to their potential impact on the population (Fig. 6), although seismic activity of any magnitude is the most frequent geological event in Tenerife. Details can be found in Table A of the Supplementary Material.

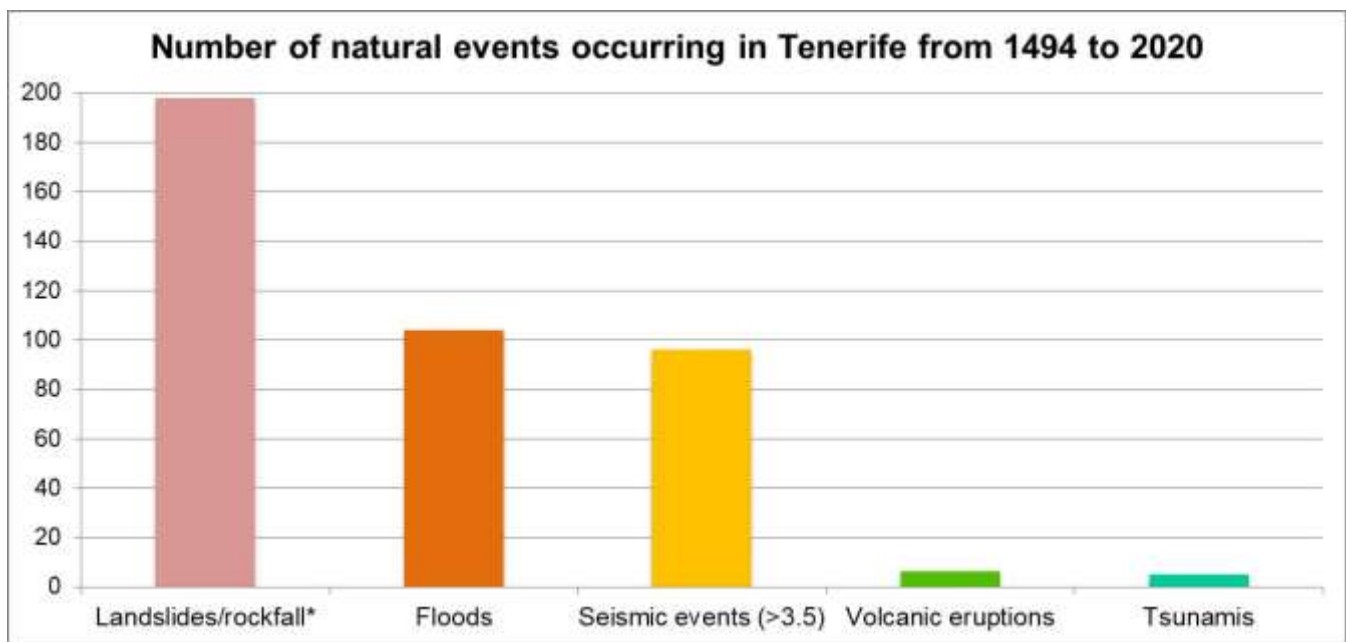


**Figure 7. Frequency of different types of natural hazards compiled in this analysis that occurred in Tenerife between 1494 and 2020. \*Note: Rockfalls/landslides display the number of events of this type occurring between 1941 and 2020 due to the lack of earlier data, hence the subdued color bar. It is important to note that only seismic events with a magnitude greater than 3.5 mbLg(L) have been considered.**

The database reveals that most landslides and rockfalls during the study period occurred along roadsides, closely linked to human activities such as road construction, which results in steep, unstable slopes. Another common location for these events is on coastal cliffs and ravines throughout the island. The data also indicate that, in some years, up to 22 such events were recorded. They generally occur more frequently in February and November, followed by January, October, and December, although they can also happen throughout the rest of the year. According to the information presented in Table A

of the Supplementary Material, these events typically occur very quickly, lasting from a few seconds to a few minutes. In some cases, instability persists for hours or days, but the actual rockfall is instantaneous. In general, the causes of these events are difficult to determine. However, most do not result in subsequent hazards, with the exception of the landslide on July 7, 1941, which was triggered by the impact of a wind wave, causing part of a cliff to collapse into the sea and generating a tsunami that inundated the coast of Santa Cruz de Tenerife (Fig. 1a).

Between 1941 and 2020, landslides and rockfalls caused a total of 16 fatalities, most of them occurring on beaches or in ravines, such as the Infierno Ravine (west of Tenerife), a popular hiking route, or in houses near slopes. Additionally, at least 10 injuries were recorded. These events generally do not require population evacuations and typically do not result in significant economic losses. Cleanup and recovery efforts in affected areas are usually quick and straightforward, with normal conditions restored within the same day or the following days.

Regarding floods, they are concentrated mainly in November and December, with less frequent occurrences in October and January, followed by February, March, and April. However, while autumn and winter floods remain relatively constant,



primarily occurring in November and December, floods in spring have shown a slight trend toward increased frequency from January to March, extending into April and May, and even sporadically occurring in August. This seasonal distribution aligns with the Atlantic storm patterns mentioned earlier. Flood events generally last between one and three days, though
some persist for up to a week. Most floods occur in the northern part of the island, coinciding with the areas of highest precipitation, as shown in Fig. 4.

Like landslides and rockfalls, floods are often triggered by a preceding event, typically heavy storms. However, unlike landslides and rockfalls, floods can themselves initiate cascading hazards. These multi-hazard scenarios are typically characterized by intense rainfall leading to flash floods, which overflow through ravines and inundate population centers in
low-lying coastal areas. Additionally, these floods are usually accompanied by significant sediment transport. To a lesser extent, some floods are also caused by wave action during storm events with strong winds.

Floods are responsible for significant human and economic losses. According to recorded events, approximately 800 fatalities were attributed to floods throughout the study period, although historical accounts suggest that the actual number may be higher. Numerous injuries and evacuations were also reported. Economic losses recorded between 2000 and 2020
alone exceeded 37 billion euros. Typically, these losses result from the destruction and damage of houses, buildings, infrastructure, and municipal facilities. It is important to note that due to technological advancements, inflation, and urban expansion, economic losses have increased in value toward the end of the study period. Available data indicate that floods pose greater challenges for emergency management and recovery than other hazard types. In earlier centuries, emergency management was primarily the responsibility of local mayors and municipal authorities, supported by the Insular Council.
Recovery efforts were largely carried out by local workers with strong community participation. However, in recent times, these tasks have been integrated into the Spanish national risk management system.

Tectonic earthquakes affecting Tenerife are also relatively frequent and typically have their epicenters in the surrounding sea. Many of these earthquakes occur in the marine area between Gran Canaria and Tenerife, over the fault zone depicted in Fig. 2. On land, seismic activity is most commonly recorded in the Icod and Orotava Valleys in the north of the island. The
strongest perceived shaking is concentrated in the northern region, with lesser effects in the metropolitan, northwestern, eastern, and southern zones. These earthquakes generally last a few seconds to a few minutes, though foreshocks and aftershocks may persist for several days before and after the mainshock.

In the context of Tenerife and the study period, tectonic earthquakes typically do not trigger additional hazards, and their immediate effects are often negligible. Occasionally, they have caused landslides and rockfalls. No fatalities were recorded
during the study period, either because they did not occur or because they were not documented. Few injuries and evacuations were reported, with panic among the population being the most common consequence. No significant damage or economic losses were documented. The highest recorded magnitude was 6.3 M (M = estimation of Mw moment magnitude in the Canary Islands based on correlations obtained in Rueda et al., 2020), though most recorded events were below 5.0 M. The highest calculated epicentral intensity was 7.3. Volcanic earthquakes, however, tend to have higher epicentral intensities



and magnitudes exceeding 6 M. Due to their characteristics, these events did not require significant management measures or recovery efforts.

Volcanic eruptions were the least frequent geological events during the study period. The first documented eruption, corroborated by geological evidence, was the Siete Fuentes-Fasnia eruption, which began on December 31, 1704, in eastern Tenerife. Seventeen days after its conclusion, the Arafo eruption began in the nearby Güímar Valley. The next eruption,

Garachico (Arenas Negras), occurred 404 days after the end of the Arafo eruption, in the northwest of the island. Between the end of this event and the start of the Chahorra (Narices del Teide) eruption in 1798, 91 years, 11 months, and 25 days elapsed. The last recorded eruption was the Chinyero event, occurring from November 18 to 27, 1909, in northwest Tenerife, 111 years, 2 months, and 3 days after the previous eruption.

All recorded eruptions took place in rift zones. The Siete Fuentes, Arafo, and Fasnia eruptions originated in the Dorsal Rift

Zone; the Arenas Negras and Chinyero eruptions in the Santiago Rift Zone; and the Narices del Teide eruption on the southwestern flank of Pico Viejo, though it remains unclear whether it should also be classified under the Santiago Rift Zone. These eruptions produced multiple associated hazards, including seismic activity, explosions, ashfall, pyroclastic ejections, lava flows, and landslides. The affected area varied depending on the hazard.

All historical eruptions in Tenerife were of the Strombolian type, with Volcanic Explosivity Index (VEI) values between 2

and 3 and lava flow volumes ranging from 0.004 km³ to 0.035 km³.

In summary, the possible scenarios that can be found in Tenerife given its historical record are as follows (Table 4):

**Table 4. Possible Natural Hazard Scenarios for Tenerife According to its Historical Record (1494-2020).**

| Outcome | Primary/direct hazards | Secondary/indirect hazards |
|---|---|---|
| Volcanic eruption | Seismicity | - |
| | | Rock falls |
| | Gases | - |
| | Explosions | - |
| | Lava flows | - |
| | | Wildfires |
| | Fallout | - |
| Earthquake | Seismic foreshocks | - |
| | Seismic aftershocks | - |
| | Rock falls | - |
| | Landslides | - |
| | No effects | - |
| Distal earthquake | Tsunami | Flooding |
| | No effects | - |
| Others (storms, human action, unknown) | Rock falls | - |
| | Landslides | - |
| | Flooding | - |



| Outcome | Primary/direct hazards | Secondary/indirect hazards |
|---|---|---|
| | | Debris flow |
| | | Erosion |
| | | Electrocution |
| | | Wildfire |
| | | Famine |
| No effects | | |

## 6 Discussion

The recorded events enabled us to compile a catalog of all possible multi-hazard scenarios that have occurred in Tenerife
over the past 526 years. Therefore, any of these events can be considered to have a probability greater than zero of recurring
in the future. However, this is not an exhaustive list of all possible scenarios for the island, as some events with much longer
recurrence periods, such as major eruptions, may have occurred before the analyzed period. Nevertheless, all the scenarios
listed here are more likely to occur in the future than any scenario not included in Table 4.

This historical data compilation allowed us to identify both successful strategies and areas for improvement in the
management of natural hazards. Based on these analyses, we extracted valuable insights and lessons from the dataset. By
identifying patterns and recurring trends, we determined common vulnerabilities, persistent challenges, and effective
resilience measures. Collectively, these findings contributed to the development of recommendations and guidelines for
future hazard preparedness, response, and recovery efforts in Tenerife. This preliminary assessment also provided insight
into the possible scenarios that could unfold on the island.

According to the records, floods are most likely to occur during the autumn-winter seasons, with the highest probability
between November and January. However, the data suggest that flooding events may also extend into the winter-spring
period, from January to May. The most probable cause of these scenarios is heavy rainfall associated with storms, leading to
the overflow of ravines. This pattern aligns with the distribution of storms throughout the year (i.e., intense waves as a cause
of flooding are less frequent than storms in Tenerife, and even less so due to tsunamis; other causes, such as landslides
obstructing the ravine course and triggering upstream flooding, are not considered due to a lack of evidence). The island's
subtropical climate, influenced by Atlantic storms, contributes to these heavy rainfall events, particularly during these
periods of the year. These storms typically approach from the ocean towards the land from the north, resulting in the heaviest
precipitation in the northern and northwestern regions of the island.

Given Tenerife's steep slopes, heavy rainfall is also expected to trigger flash floods—sudden and rapid flooding events
characterized by swift-moving water, which can be particularly hazardous due to their speed and the limited time available
for evacuation or preparation. The main risk associated with these events is the overflow of ravines, which typically occurs
when they reach flatter areas or zones modified by human activity, such as channeling, bridge construction, flow alterations,
or paving. The most affected municipalities include Santa Cruz de Tenerife, San Cristóbal de La Laguna, Garachico



(particularly by waves), San Andrés, La Orotava Valley, Güímar, Puerto de la Cruz, Los Realejos, and Icod de los Vinos,
among others (Fig. 1a). Additionally, torrential floods transport large volumes of sediment through ravines, and when
combined with landslides occurring simultaneously, they result in debris flows that deposit mud and large rocks in overflow
areas. This exacerbates the impact and complicates cleanup and recovery efforts.

Overall, the frequency of this phenomenon has remained relatively constant throughout the study period, but a slight increase
has been observed in recent years. However, its impact has clearly intensified, with flooding events affecting metropolitan
and coastal areas more frequently. The findings indicate that this scenario causes the most damage and has the potential to
become even more severe in the future. The distribution of major population centers is a critical factor, as most are situated
in areas with high rainfall, often along or near major ravines. Many of these ravines have been paved and converted into
streets, while others have been modified with infrastructure that disrupts natural drainage toward the sea. It is crucial to
recognize that water will always follow its natural course, meaning that reactive damage repairs, rather than proactive risk
mitigation measures, are likely to have greater economic and social consequences.

To mitigate flood risks, the following measures should be considered:

1. Clearing ravines to reduce solid load during torrential floods.
2. Creating water retention areas, such as parks, gardens, or other open, permeable spaces to accommodate controlled water overflow.
3. Reforesting headwaters and high-precipitation areas to reduce runoff and mitigate wildfire risk through improved forest management and removal of combustible materials.
4. Increasing channel permeability by avoiding paved ravine beds.
5. Raising the height of bridges crossing ravines to prevent obstruction during floods.
6. Improving drainage toward the sea by cleaning and expanding the sewage system, opening its outlet to the beach while protecting it from sand blockages during storms, and increasing the slope toward the sea to prevent wave intrusion, taking into account projected sea-level rise (Fig. 3).

Preserving the natural alignment of ravines and integrating land-use planning accordingly.

All these measures, along with others, should be designed based on detailed flood hazard mapping, taking into account
potential secondary risks arising from their implementation.

Another common hazard in Tenerife is rockfalls and small-scale landslides. These typically occur along road cuts or within
ravines, the latter being less frequented by tourists. Consequently, such events generally do not result in fatalities, injuries,
significant damage, or high costs. However, notable exceptions include the landslide in Barranco del Infierno on October 26,
2015, which resulted in one fatality and four injuries, and the rockfall on November 1, 2009, in Los Gigantes (west coast),
which caused the deaths of two people at Santiago del Teide beach. Another significant landslide occurred on January 27,
1947, in Tacoronte, resulting in five fatalities (IGME, 2016).

It is worth noting that historical records likely underestimate the occurrence of such events, as they appear less frequently
documented compared to other hazard types. This limitation could not be fully addressed, as it is not possible to assume a





constant frequency over time; the occurrence of landslides depends on multiple factors distinct from those influencing other hazards. For example, road construction can increase slope failures, while variations in rainfall and other environmental

factors also play a role. However, while it cannot be assumed that past frequency was significantly lower, records indicate that despite the high frequency of these events in recent years, they have not caused substantial economic or social losses, suggesting they may not represent a critical future threat. Nevertheless, their consequences, such as road closures and risks to individuals, should not be overlooked.

Despite uncertainties in the data, records indicate that rockfalls and landslides result from various factors, including

temperature fluctuations, rainfall, and earthquakes acting on slopes in combination with gravity. Most events occur along roads or in areas modified by human activity. In some cases, landslides follow periods of rainfall, earthquakes, floods, strong winds, or intense waves. However, their occurrence is more evenly distributed throughout the year than floods, making it difficult to establish a single triggering cause. Some events have been preceded by earthquakes, which may have destabilized slopes, leading to landslides in subsequent days.

Given this, preparedness measures are essential, particularly after heavy rainfall and seismic activity. Recommended actions include:

- Modifying the slopes of roadsides prone to landslides.
- Installing protective structures such as meshing and bolting to retain small landslides.
- Removing unstable rock masses from road embankments.

- Conducting thorough geological and geotechnical studies before creating new road cuts.
- Preserving the original topography as much as possible to avoid the creation of excessively steep slopes, which, despite their practicality in volcanic terrain, may lead to long-term consequences.

Earthquakes felt in Tenerife generally originate from either tectonic or magmatic sources. Tectonic earthquakes are typically associated with the transform fault running through the channel between Gran Canaria and Tenerife. This area exhibits the

highest seismic activity in the Canary Islands, with 400–500 earthquakes per year below 2.5 mbLg, of which fewer than a dozen exceed this magnitude (Instituto Geográfico Nacional, 2022). Earthquakes recorded in Tenerife have reached maximum intensities of 1–6 on the MSK-64 scale (Medvedev & Sponheuer, 1964) and rarely cause damage. Notable exceptions include the May 9, 1989 earthquake (M5.0), which had a maximum epicentral intensity of 6.7, causing minor structural damage (Mezcua et al., 1992).

Magmatic earthquakes, often occurring in swarms, have not caused significant damage in Tenerife. Other hazards affecting the island include Saharan dust storms, wildfires, and tsunamis, though the latter are rare.

Other hazards affecting Tenerife include haze originating from the Sahara, which, in addition to causing respiratory problems, has occasionally been accompanied by locust plagues, as occurred in 2004. Forest fires, often triggered by heat waves and/or human activity, represent another significant hazard. Notably, the 2007 wildfire burned approximately 15,000

hectares—an area that was once again affected in the 2023 fire. The presence of haze can further exacerbate wildfire conditions. Additionally, while tsunamis are less common, historical records indicate the occurrence of a tsunami on March



31, 1761. However, some of these records are not well substantiated. Nevertheless, as an island located in the middle of the Atlantic Ocean, Tenerife remains susceptible to any tsunami that propagates through this region.

To further enhance our understanding and improve the precision and accuracy of the data used in this study, it is crucial to conduct additional fieldwork and integrate citizen science initiatives. This roadmap highlights the necessity of comprehensive geological studies through field campaigns in Tenerife and underscores the importance of investing in Earth sciences and fundamental geological research, particularly in the field of physical geology. Fieldwork plays a pivotal role in scientific research, especially in geological studies. Direct observation and data collection in the field provide critical insights into geological processes and phenomena occurring within a given region. In the case of Tenerife, conducting field campaigns will yield invaluable information on the island's geological characteristics, including its volcanic activity, tectonic processes, and hazard-prone areas. By supplementing our dataset with new field observations, we can refine our understanding of the local geology and enhance the accuracy of our analyses.

Moreover, incorporating citizen science initiatives can significantly contribute to the acquisition of valuable data. Citizen science involves engaging the public in scientific research, enabling individuals from diverse backgrounds to actively participate in data collection and analysis. In Tenerife and other regions, citizen scientists can assist by reporting observations, sharing photographs, and providing local knowledge of geological events and processes. Community involvement facilitates the collection of a broader range of data points, improves spatial coverage, and enhances our comprehension of the region's geological dynamics.

To effectively implement citizen science initiatives, it is essential to establish collaborative platforms and communication channels between scientists and the public. These platforms can streamline data sharing, provide standardized data collection protocols, and ensure the accuracy and reliability of the collected information. Engaging citizens in the scientific process not only expands the available dataset but also fosters public awareness and understanding of geological hazards and their potential impacts.

## 7 Concluding remarks

The creation of a comprehensive multi-hazard database for Tenerife, encompassing events from 1494 to 2020, represents a pivotal step toward understanding and preparing for the diverse challenges faced by the island. Despite acknowledging the inherent uncertainties in the data, this catalog serves as a valuable resource for long-term multi-hazard assessments. The uncertainties highlighted in the discussion underscore the need for continuous research and fieldwork to refine our understanding.

Such databases play a critical role in complementing short-term hazard assessments, guiding the design of monitoring networks, and informing the development of effective risk mitigation strategies. By fostering a deeper understanding of historical events, these databases contribute to the formulation and implementation of disaster risk reduction policies. They

serve as a cornerstone in strengthening regional risk management systems, enhancing resilience against a complex array of hazards.

The utility of this database extends beyond academic research, offering practical applications for regional policymakers and risk management authorities. As we navigate the uncertainties associated with natural hazards, the establishment of such databases stands as a proactive measure, providing a foundation for informed decision-making and fostering a resilient and adaptive response to the challenges posed by Tenerife's dynamic geological landscape.

**Data availability**

The dataset supporting the findings of this study is publicly available through the digital repository Digital.CSIC. The dataset can be accessed directly via the following DOI: https://doi.org/10.20350/digitalCSIC/17088 and should be cited as López-Saavedra et al., 2025. No additional access tokens or review links are required.

**Author contribution**

MLS: Conceptualization, Data curation, Formal analysis, Funding acquisition, Investigation, Methodology, Visualization, 515 Writing – original draft preparation; JM: Conceptualization, Funding acquisition, Methodology, Project administration, Resources, Supervision, Validation, Writing – review & editing; MMS: Data curation, Formal analysis, Methodology, Software.

**Competing interests**

The authors declare that they have no conflict of interest.

**Aknowledgements**

This work was partially supported by the EG grant EVE (DG ECHO H2020 Ref. 826292) and the CSIC grant MAPCAN (CSIC Ref. 202130E083). Marta López-Saavedra received an FPU PhD grant (FPU19/02413) from the Ministry of Science, Innovation and Universities of the Government of Spain. We thank the projection authors for developing and making the sea-level rise projections available, multiple funding agencies for supporting the development of the projections, and the 525 NASA Sea-Level Change Team for developing and hosting the IPCC AR6 Sea-Level Projection Tool. We also thank the collaboration and dedication of Carlos Rodríguez, from the Provincial Historical Archive of Santa Cruz de Tenerife.



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
