# Peer review of "Creation and analysis of a multi-hazard database: Tenerife (Canary Islands) as a case study"

_Earth System Science Data, 2025_

## Referee Comment (RC1)

The manuscript titled "Creation and analysis of a multi-hazard database: Tenerife (Canary Islands) as a case study" presents a timely and commendable effort to enhance our understanding of historical natural hazards in island environments. By compiling and analyzing over five centuries of hazard events—including volcanic eruptions, earthquakes, landslides, floods, and tsunamis—the authors contribute to the historical evidence base necessary for improving disaster risk reduction (DRR) planning in volcanic and coastal regions.

The creation of this multi-hazard database represents a meaningful step toward integrated risk assessment, particularly in contexts such as Tenerife, where geographic isolation, complex geomorphology, and socio-economic exposure shape distinct patterns of vulnerability. The dataset, provided as supplementary material, offers valuable insights into hazard frequency, spatial distribution, and early forms of risk management. These insights could support scenario planning and inform both retrospective and prospective DRR strategies.

The authors' core contribution lies in proposing a systematic methodology for compiling historical hazard records and analyzing their temporal and spatial dynamics. They suggest that the dataset can help identify patterns of vulnerability and response, offering a foundation for improved resilience and even applications in risk modelling and machine learning. The goal is to create a reference dataset capable of bridging the gap between past hazard knowledge and future-oriented, systemic risk governance frameworks. While this objective is both relevant and promising, several aspects of the methodology, conceptual framing, and practical applicability merit further scrutiny. In particular, the lack of standardized classification criteria, limited integration of hazard interdependencies, and insufficient engagement with dynamic, multi-risk frameworks constrain the potential of the dataset to fulfill its broader claims.

The analysis described in the paper often falls short of the rich information provided by the dataset. Indeed, an effort to provide a deeper understanding of frequency, seasonality or spatial distribution could be undertaken. Note, for example, that Figure 2 counts the number of multi-hazard events considered in the dataset with simple frequency, comparing low frequency-high damaging events, such as volcanic eruptions, with more common and less damaging events such as landslides-rockslides. In this respect, it is suggested that the figure is substantially improved to provide a deeper understanding of the event set.

The authors describe the dataset as a systematic collection of "types of events." However, the current structure contributes to some ambiguity, particularly in how events, hazards, and impacts are classified. Although the event typology appears closely linked to available sources and official reporting, the lack of alignment with standard classification systems—such as the EM-DAT disaster typology (Guha-Sapir et al., 2017)—and the absence of clear definitions or categorization logic undermine the dataset's potential usability. These limitations reduce its applicability for comparative analyses, probabilistic risk modelling, or integration with early warning systems and DRR strategies, as envisioned in the manuscript's introduction. While future iterations of the database may focus on harmonization and interoperability, the current version would benefit from a clear justification of the classification choices made, an acknowledgment of their limitations, and a discussion on how they might be refined or aligned with existing standards in the future.

The dataset accounts mainly for geological events such as earthquakes, volcanic eruptions, landslides and seisms, but authors integrate flooding as a geological event without further justification. Flooding is systematically considered an hydrological hazard, not a geological one. Moreover, the list of natural risks for Tenerife in Table 3, classifies Floods within hydrological risks. Therefore, a proper definition of flood throughout the text, particularly with respect to additional terms such as torrential floods and flash floods used to characterize the type of flooding occurring in steep islands in the Macaronesian region, and justification to include it as a geological hazard in the dataset is suggested.

While the authors suggest that the event set can support the understanding of multi-hazard dynamic risk, the analytical narrative could improve to contribute to this objective. Specifically, the terms multi-hazard, event type, and impact are not clearly defined or consistently used throughout the text. As a result, it remains difficult to assess the extent to which the database captures interrelations between hazards or contributes to a systemic understanding of risk dynamics in the context of hazards combinations. In its current form, the paper makes a valuable empirical contribution, and the dataset can certainly support future research on multi-hazard risk and system-level vulnerability. However, to fully realize this potential, the authors could better clarify the terminology used (hazard, event, impact) and outline possible future steps to categorize hazard relationships more precisely, linking their approach more explicitly to existing multi-hazard or risk governance frameworks in subsequent work.

Section 2: Geological context and natural hazards

Section 2 offers a useful geological overview of the Canary Islands, focusing on the formation and volcanic history of Tenerife. While this helps set the stage for understanding the island's hazard profile, the section feels somewhat insular in scope. It is striking that the geological narrative does not reference other significant volcanic events in the archipelago—most notably, the 2021 Tajogaite eruption in La Palma, which stands as the most damaging natural disaster in the Canary Islands' recorded history.

By omitting this broader perspective, the section inadvertently conveys a sense of volcanic risk as unique to Tenerife. This weakens the opportunity to contextualize the region's multi-hazard dynamics, and the systemic nature of volcanic risk across the region. Including a brief mention of past and recent eruptions beyond Tenerife (La Palma at least) would help frame the database as part of a regional risk landscape and contribute to a more integrated understanding of hazard interconnections.

In addition, the section could be strengthened by reflecting more directly on the types of geological interactions captured in the dataset. Even a preliminary discussion of how different hazard types (e.g., seismicity, landslides, eruptions) may have interacted historically would provide a useful bridge between the physical setting and the subsequent data analysis. Although the database covers events up to 2020, it may be worth briefly acknowledging the recent uptick in seismic activity, which reinforces the contemporary relevance of the database and suggests avenues for its future update and use.

Section 3. Natural hazards, risk management and risk regulation

Section 3 presents several disconnected elements that would benefit from clearer purpose and structure. The first part of the section is devoted to replicating the methodology used by the Cabildo de Tenerife's PEIN (2020) to assess natural risk levels through an index based on hazard probability (0–5) and risk severity (0, 1, 2, 5, 10). The combination of these scores

generates an overall risk score from 0 to 50 and classifies severity from Low to Very High, later applied to a list of hazards.

While this description may be useful in explaining how risk is formally assessed at the local level, it appears tangential to the core research aims of the paper. The authors do not justify the relevance of including this methodology, nor do they provide scientific references or a critical discussion of its assumptions or limitations. A more concise explanation, alongside Table 3, would likely suffice. In its current form, this section adds a layer of institutional detail without showing how this risk classification contributes to the construction or interpretation of the historical hazard database. This disconnection introduces a sense of subjectivity in the selection or weighting of hazard-event relationships.

The second part of the section shifts focus to future climate change scenarios in the Canary Islands—particularly increasing temperature, sea level rise, and changing precipitation patterns. However, the narrative only introduces the current weather regimes of Tenerife toward the end of the section, along with historical precipitation and temperature maps. This order weakens the analytical flow: it would be more logical to first describe the island's present-day meteorological patterns and hydrometeorological risk drivers, and then introduce how climate change is projected to alter these conditions. Such a structure would help readers understand how ongoing and future climate shifts could affect the frequency and severity of extreme events—and why existing risk classifications may require updating.

In light of these observations, the authors could consider restructuring this section by:

- Beginning with the description of Tenerife's climate conditions and weather types;
- Following with climate change scenarios and their implications for hazard patterns;
- And moving the explanation of the PEIN risk index and Table 3 to a dedicated section on disaster risk reduction (DRR) planning, where its institutional role can be better contextualized.

Finally, the section would benefit from a brief overview of the overall risk and emergency coordination structure in the Canary Islands. While the PEIN is a key territorial risk plan for Tenerife, it exists within a broader governance architecture that includes specific emergency response plans for each type of hazard (volcanic, flood, wildfire, etc.). This fragmentation—based on a single-risk planning model—is not acknowledged in the paper, and neither is the lack of integrated multi-hazard emergency coordination or communication frameworks. Addressing this gap, even briefly, would strengthen the authors' position and reinforce the relevance of a historical multi-hazard dataset for improving future DRR strategies.

Section 4. Methodology

The Methodology section outlines the process by which the historical hazard database was compiled. While the authors present a structured workflow, there are two key issues that deserve attention.

As mentioned before, the classification of hazards appears to contain some inconsistencies. Notably, floods are categorized as geological hazards, which contradicts widely accepted definitions used in international and national risk frameworks (e.g., UNDRR, EU Risk Typologies, Spanish Civil Protection legislation). Floods are typically considered hydrometeorological hazards, with geological hazards referring to earth-origin phenomena such as landslides, earthquakes, or volcanic eruptions. This classification should be further

justified and suggests a need for greater alignment with standardized hazard typologies, particularly if the dataset is to serve comparative or modeling purposes beyond the regional context.

On the other hand, while the authors refer to a systematic approach to event identification and classification, the methodology lacks references to established multi-hazard or disaster risk frameworks, which would lend robustness and replicability to their approach. For instance, the manuscript does not draw on available taxonomies or concepts distinguishing: primary vs. secondary hazards, triggering relationships (e.g., rainfall-induced landslides), or compound, cascading, and interacting hazard events. The manuscript would benefit from anchoring its methodology in existing risk classification standards and definitions, as provided by international scientific literature and operational frameworks. For example, López-Saavedra and Martí (2023) offer a structured review of conceptual distinctions between hazard, risk, and impact, as well as the types of interactions (e.g., compound, cascading, or interconnected hazards) that are critical in multi-hazard risk assessment.

While the methodology is clearly structured and reflects a commendable effort to organize diverse historical records, it would benefit from greater alignment with existing frameworks in the field of multi-hazard risk analysis. At present, the paper does not reference widely used classification systems or conceptual approaches developed by organizations such as the UNDRR, or key academic contributions (e.g., Gill & Malamud, 2016; Kappes et al., 2012; de Ruiter & van Loon, 2022). Integrating—or at least situating—the proposed classification in relation to such frameworks could enhance both the transparency and transferability of the work. Given the authors' stated aim to contribute to understanding dynamic, multi-hazard risk, a brief explanation of the rationale behind their categorization choices, and how these align with or diverge from established standards, would help clarify the dataset's scope and improve its broader analytical and policy relevance. Such reflection could also open the door to future development of the database toward greater interoperability and utility across different risk governance contexts and regions.

Section 5. Results

Section 4 presents the results of the event compilation and hazard analysis, but would benefit from greater internal consistency, clearer references, and stronger integration with earlier sections of the paper.

To begin with, there appear to be figure reference inconsistencies that should be corrected for clarity. For example:

- Line 296 refers to Figure 6 while describing volcanic eruptions; it seems the correct figure might be Figure 2.
- Line 303 again cites Figure 6 for seismic activity, when this likely corresponds to Figure 7.

In Line 303, the authors also claim that seismic activity is the most frequent hazard in Tenerife if earthquakes below 3.5 mbLg are considered. This statement could be strengthened by referencing a reliable data source—such as the Instituto Geográfico Nacional (IGN)—to support the frequency threshold used.

As previously mentioned, Figure 7 presents a useful summary of seismic activity, but it could benefit from the richness of the dataset. The authors could enhance its value by incorporating additional dimensions in the analysis, such as a timeline distribution of events,

seasonality, or disaggregation by magnitude and location. Separate figures by hazard type could also allow for better visualization of trends and interactions.

In Line 310, the authors note that landslides and rockfalls occur mainly on roadsides and are associated with human activities. However, this framing risks underestimating unreported or undocumented events in uninhabited or infrastructure-free areas. In contexts where no direct damage is recorded, such events may remain absent from historical datasets. This illustrates a broader issue in the paper, where hazard and risk concepts are sometimes used interchangeably. In this case, the authors might take the opportunity to qualify their statements more carefully and acknowledge the limitations of the dataset in fully capturing the dynamics of hazard occurrence and impact.

Relatedly, in Line 320, the authors report a total of 18 fatalities in beaches and ravines associated with landslides and rockfalls. They then state that such events do not result in significant economic losses or require population evacuation, and that recovery is typically straightforward. These conclusions may be overly confident given the likely limitations of available records. For example, some of these events may have caused prolonged closures of beaches or trekking areas, with economic and reputational effects on tourism-dependent municipalities. In some cases, victims were tourists, which raises questions about impacts on risk perception, insurance, and emergency response capacity. Without documented evidence on compensation, loss of income, or indirect effects, it would be advisable to present these conclusions with greater caution.

In Line 317, the phrase "However, most do not result in subsequent hazards…" is somewhat unclear. Since the paper does not previously discuss hazard interconnections or cascading effects, this sentence may be confusing to readers. The authors could consider introducing relevant definitions of hazard interactions earlier in the text—possibly drawing on sources such as López-Saavedra & Martí (2023) or the Handbook on Multi-Hazard, Multi-Risk Definitions and Concepts (Ward et al., 2022). This would provide a useful conceptual basis for interpreting their findings.

In Lines 329–332, the manuscript refers to storm patterns linked to flooding events, but this link is not clearly developed in earlier sections. For instance, the role of tropical-like storms and Atlantic squalls is mentioned in Section 3 but not sufficiently explained. Revisiting this earlier description and clarifying how such meteorological systems relate to compound hazards, especially floods, may enhance both the internal coherence of the manuscript and its contribution to understanding multi-hazard dynamics in the region.

The reference to "precedent events" in Line 332 could benefit from greater clarity, especially if the authors intend to describe causal or sequential hazard relationships. In most cases, floods in the Canary Islands are the result of meteorological triggers such as heavy rainfall or storm surges, and are thus typically classified as primary hydrometeorological hazards. However, the authors may wish to clarify whether they refer to purely meteorological drivers or also consider flood events that arise from compound or cascading processes (e.g., landslide dam failures or volcanic triggers). Making these distinctions clearer would help improve the conceptual consistency of the hazard classifications used in the paper.

Section 6. Discussion

Lines 384–389. The authors conclude that sufficient insights have been extracted from the historical hazard dataset to inform Disaster Risk Reduction (DRR) recommendations, specifically offering seven measures focused on flood risk reduction. However, there

appears to be some disconnect between the descriptive findings of the database and the proposed recommendations. First, the discussion could benefit from clearer consideration of multi-hazard and dynamic risk contexts, particularly within complex flood scenarios. For example, how might each of the proposed measures interact with other hazards—either reinforcing or undermining risk reduction goals? A more explicit exploration of synergies and trade-offs would improve the strategic coherence of the proposed actions.

Second, the recommendations could be strengthened by drawing on evidence-based literature or case studies demonstrating the effectiveness of such measures in similar geographic or institutional contexts. In their current form, the proposals read more as general suggestions than conclusions directly derived from the dataset analysis or the local policy landscape.

Third, the discussion omits key elements of existing flood risk planning frameworks in Tenerife. Royal Decree 903/2010 of 9 July, which transposes the EU Floods Directive (2007/60/EC), establishes Spain's formal framework for flood risk management. This includes preliminary risk assessments, flood hazard and risk mapping, and the development of Flood Risk Management Plans (FRMPs). Tenerife has already completed three full planning cycles (2015–2021, 2021–2027, and the ongoing 2027–2033), and many technical and institutional measures might be already embedded in these plans. Therefore, the paper could at least refer to the proposed DRR measures with existing planning instruments, noting areas of alignment or divergence. This would also be an opportunity to examine whether current official planning processes are integrating—or neglecting—multi-hazard scenarios, especially given the multi-risk ambition of the database.

Lines 426–427. There appears to be a referencing inconsistency: Figure 3 is cited, but the correct figure may be Figure 4. Additionally, Line 427 seems to describe Measure 7, though this is not clearly indicated.

Line 450. The final set of five landslide risk reduction measures is presented without supporting evidence or references. Providing links to empirical examples or planning experiences would bolster the credibility of these suggestions. In this context, it might be valuable to acknowledge the role of path-dependent adaptation—that is, how past decisions and institutional learning have shaped current risk management strategies in Tenerife or in the Canary Islands. This could reveal both limitations and opportunities for transformative change.

Lines 467–471. The brief mention of other hazards—such as haze episodes, droughts, and heatwaves—deserves further development. These meteorological conditions are well-known triggers of wildfires, especially when compounded by human activity, and the manuscript would benefit from elaborating on these interactions. For example, the 2007 and 2023 wildfires are compared based on the surface area burned, yet the phrase "the same area affected" could be misinterpreted as affecting the same geographic area: rephrase by "number of hectares affected". Clarifying this distinction is important. Moreover, the potential cascading effects of such hazards—e.g., increased erosion, biodiversity loss (including damage to the laurel forest), and downstream flood risk—could be more thoroughly analyzed, especially given the database's multi-risk potential.

Lines 475–end. The final section suggests that citizen science initiatives may improve the forensic analysis of past hazard events through collaborative platforms and public engagement. This is an important and promising avenue. However, it could be reconciled

with the more ambitious expectations set in the Introduction, where the authors claim the dataset could support risk modelling and machine learning applications (such as Classen et al., 2023). In its current state, the dataset lacks sufficient metadata standardization, clear indicators, validation protocols, and methodological documentation to support such advanced uses. These limitations should be acknowledged explicitly, particularly regarding the representativeness, reliability, and interoperability of the dataset. A transparent discussion of these constraints would not only improve the manuscript's credibility but also help future researchers understand the scope and current limitations of this valuable resource.

A final comment on integrating vulnerability and socio-economic dynamics to strengthen risk analysis:

Throughout the manuscript, the authors offer a rich historical overview of hazard events in Tenerife, but the analysis remains largely hazard-centric, with limited attention to the underlying social, economic, or institutional drivers of vulnerability. Over the 400-year period covered, Tenerife island and the region has undergone profound transformations—including population growth, tourism development, land-use change, and infrastructure expansion—all of which deeply shape risk exposure and, ultimately, adaptive capacity. A brief description or a discussion of how these processes have evolved and influenced vulnerability over time, seems like a great opportunity to contextualize the data and offer a more systemic understanding of risk transformation. Acknowledging these socio-economic dynamics would help position the dataset as a more holistic tool for future risk assessments, and align the work more closely with contemporary approaches in disaster risk science that emphasize the co-evolution of hazards, exposure, and vulnerability.

References:

de Ruiter, M. C., & van Loon, A. F. (2022). A typology of compound weather and climate events. Nature Reviews Earth & Environment, 3(5), 333–347. https://doi.org/10.1038/s43017-022-00275-5

Gill, J. C., & Malamud, B. D. (2016). Hazard interactions and interaction networks (cascades) within multi-hazard methodologies. Earth System Dynamics, 7(3), 659–679. https://doi.org/10.5194/esd-7-659-2016

Guha-Sapir, D., Hoyois, P., Wallemacq, P., & Below, R. (2017). EM-DAT: The Emergency Events Database – Technical Report 2000–2019. Centre for Research on the Epidemiology of Disasters (CRED). https://www.emdat.be

Kappes, M. S., Keiler, M., von Elverfeldt, K., & Glade, T. (2012). Challenges of analyzing multi-hazard risk: A review. Natural Hazards, 64, 1925–1958. https://doi.org/10.1007/s11069-012-0294-2

López-Saavedra, L., & Martí, J. (2023). Revisiting multi-hazard and multi-risk assessment frameworks: Definitions, concepts and applications. International Journal of Disaster Risk Reduction, 93, 103705. https://doi.org/10.1016/j.ijdrr.2023.103705

Ward, P. J., de Ruiter, M. C., Mård, J., Schröter, K., Van den Homberg, M., & Aerts, J. C. J. H. (2022). Handbook of multi-hazard, multi-risk definitions and concepts. MYRIAD-EU Deliverable D1.1. https://www.myriadproject.eu/resources/

Classen, M., de Ruiter, M. C., Ward, P. J., et al. (2023). Using event-based data to understand multi-hazard impacts for machine learning applications. Natural Hazards and Earth System Sciences Discussions. https://doi.org/10.5194/nhess-2023-131

---

## Referee Comment (RC2)

[referee-annotated manuscript omitted]

---

## Author Comment (AC1)

**Author comment to Julia Crummy (Referee 2)**

Dear Editor and Reviewer Julia Crummy,

I would like to express my sincere gratitude to Reviewer Julia Crummy for her thoughtful, constructive, and encouraging feedback on our manuscript. Her comments reflect a deep understanding of the topic and a commitment to strengthening the clarity, utility, and rigor of our work. We appreciate her recognition of the value of the dataset and methodology presented, as well as the manuscript's relevance to disaster risk management beyond the specific case of Tenerife.

Below, I provide a detailed, point-by-point response to each of the reviewer's comments. Changes made in response to the suggestions have been incorporated into the revised manuscript and are tracked accordingly. For clarity, each reviewer comment is restated, followed by our response.

**General Comments**

We are very pleased that the reviewer found the manuscript to be clearly written and the dataset well-structured, and we especially value the recognition of its wider applicability to other volcanic islands and contexts. We agree entirely with the importance of transparent multi-hazard analysis, and this manuscript aims to contribute toward the standardization and systematic use of such data for decision-making. The minor revisions suggested are welcome and have been addressed in full.

**Line 1** – *I would argue that it is not a database, but rather, a dataset.* - We agree with the reviewer's distinction. The manuscript now uses the term "dataset" throughout to describe the downloadable Excel file, as it is a static, curated collection of event records rather than a dynamic, relational database.

**Line 14** – *Having read through the manuscript, vulnerabilities haven't really been covered here.* – We appreciate this observation and would like to clarify the approach taken. While the manuscript does not perform a formal vulnerability assessment in the quantitative or model-based sense, the dataset captures qualitative expressions of vulnerability through documented economic losses, social impacts (e.g. , deaths, injuries, displacements), and environmental damage. These fields, together with entries related to emergency response and recovery, provide a multi-dimensional picture of societal fragility and resilience in the face of hazards. To better reflect this, we have modified the terminology in key parts of the manuscript to emphasize that we describe "impacts", which are valuable indicators of vulnerability. A full vulnerability analysis remains outside the scope of this dataset-focused paper but is facilitated by the data provided for future research. This clarification has been added in the Methods and Discussion sections.

**Line 34** – *forecasting – many of these hazards cannot be predicted.* – Agreed, the word has been corrected.

**Line 36** – *quantitative risk assessments.* – Agreed, the word has been added.

**Line 43** – *forecast.* – Agreed, the word has been corrected.

**Line 54** – *quantitative risk models. In the absence of reliable data, qualitative risk assessments can be done. The level of accuracy could be debated, but I think they are not inaccurate…* – Agreed, the word has been added.

**Line 62** – *Agreed, but robust exposure, vulnerability (physical, social, economic) and resilience data are also needed.* – Fully agreed. We have expanded this sentence to highlight the essential role of exposure, vulnerability, and resilience data.

**Line 64** – *I disagree with this – the full spectrum of impacts will depend heavily on exposure and vulnerability, which is dynamic and therefore very complex to quantify and measure. Past multi-hazard events will only give so much information and should be complemented with impacts studies, vulnerability, exposure etc data.* – The sentence has been reworded to reflect that while historical records provide a foundation, they must be complemented with dynamic data on exposure and vulnerability,

**Line 69** – *more accurate.* – Agreed, the word has been added.

**Line 89** – *Is this still debated or is it agreed that it is a mantle plume? Would it be better to state mantle hotspot?* – Unfortunately, this is still a matter of debate and it seems that it will still take quite long to reach a consensus on this issue. This is why in this paper we have preferred to not insist on it as it is not a relevant aspect for the purpose of our study. Anyway, we have slightly modified the text to clarify this point. The origin of Canarian magmatism, whether it is the result of a persistent mantle plume or active tectonics or a combination of both, is still a matter of considerable debate (Hernández-Pacheco 90 & Ibarrola, 1973; Anguita & Hernán, 1975, 2000; Schmincke, 1982; Araña & Ortiz, 1991; Hoernle & Schmincke, 1993; Carracedo et al., 1998; Fullea et al., 2015).

**Line 112** – *such as…* – Expanded to provide specific examples for clarity about different volcanic hazards.

**Line 148** – *delete (you're talking about hazards here, not risks).* – Done.

**Line 154** – *I would add "The PEIN" before "Risk assessment" here, as risk assessments usually include all components of risk not just the two parameters used here. I would argue that what they do is not a comprehensive risk assessment as, given what is described here, they do not include exposure, vulnerability or resilience.* – Done. The sentence has been corrected to "The PEIN risk assessment…".

**Line 155** – *The use of an index is normally considered a qualitative or semi-quantitative approach. I would just delete "quantitatively" here.* – Done.

**Line 186** – *This image is fuzzy for me.* – The image has been replaced.

**Line 234** – *Just wondering why drought isn't specifically included? This hazard has been a serious problem for Tenerife in recent years, and will surely worsen with the changing climate. Also, it is important in terms of multi-hazard cascades with resulting forest fires.* – Thank you for this pertinent comment. Drought is indeed a critical and increasingly relevant hazard for Tenerife, particularly given its cascading relationship with forest fires and water stress. In this section, we had listed a few illustrative hazards rather than presenting an exhaustive inventory. Nonetheless, in response to your suggestion, we have now explicitly added drought,

alongside other relevant underrepresented hazards such as heatwaves and ocean wave impact, to better reflect the full multi-hazard spectrum affecting the island.

**Line 243** – *repetition, not needed.* – Deleted.

**Line 252** – *Was it just cascading, or were coincident and compounding hazards also considered?* – This is a key distinction, and we appreciate the opportunity to clarify it. The dataset, as presented, is primarily descriptive and compiled from documented historical events, with an emphasis on capturing direct cascading relationships (i.e. secondary hazards triggered by a primary one). At this stage, we have not explicitly classified or interpreted events as "coincident" or "compounding", given the objective was to provide a reliable, objective record of events, rather than to perform a systems-based analysis of interactions. However, this classification is indeed possible using the dataset, and we are currently exploring such typologies in follow-up work.

**Line 374** – *Needs the reference: Newhall and Self, 1982.* – Citation added. Thank you for this correction.

**Line 377** – *urban fires.* – We respectfully clarify that the fires described occurred predominantly in forest contexts, especially in canarian pine forests, and not in urban areas. We have corrected the term from "wildfires" to "forest fires" to more accurately describe the events.

**Line 377** – *These should also be included with ballistics.* – We understand your concern, but our dataset includes only those scenarios supported by historical evidence for the studied time range (1496–2020). No documented cases in Tenerife during this period link volcanic ballistics with subsequent fires. While we agree such scenarios are theoretically plausible and may occur in many volcanic scenarios, including in Tenerife, their inclusion here would be speculative and outside the empirical scope of this study. Nonetheless, the revised Discussion section now clarifies that this dataset does not capture all possible hazard interactions, but rather those that are documented, with room for expanding future scenario modelling based on additional literature and regional analogues.

**Line 379** – *Not all possible. This would be assuming that the record you have is 100% complete and that what happened in the past is what will happen in the future. There are other events that can happen in the future that may have never happened before here. You talk to this further down, so I suggest just rewording this.* – Totally agreed, we have reworded to acknowledge the limits of historical completeness, and the fact that future scenarios may differ, especially under changing environmental. However, some explanations is given below that sentence.

**Line 382** – *also under-reporting, missing records etc.* – We fully agree that under-reporting and missing records are critical limitations in most historical hazard datasets. However, in the specific case referenced (volcanic eruptions on Tenerife since 1496), the documented events have been corroborated by stratigraphic and petrological studies, giving us confidence that no historical eruptions during this period have been missed. That said, we acknowledge that this level of certainty does not apply equally across all hazard types (e.g., landslides, floods). However, that statement referred only to volcanic eruptions.

**Line 383** – *I disagree with this statement. Just because something has happened before doesn't mean it will happen again. This statement might be true for some of the events you have listed*

*(e.g. flooding) but with the changing climate, other multi-hazard scenarios not listed here, might well be more likely in the future.* – We agree with the reviewer that recurrence does not imply certainty, especially in a changing climate. Our intention was never to imply determinism, but to stress that historically recurring scenarios can inform probabilistic planning. We have revised the sentence to clarify that the dataset enables the identification of plausible, historically grounded scenarios, while also acknowledging that emerging hazards and changing system dynamics may generate new or previously unrecorded interactions.

**Line 384** – *There is no inclusion of this in the text – it would be nice to see some sort of discussion on risk management.* – We appreciate this comment and recognize that more could be said. The scope of this paper is centered on the creation and structure of the multi-hazard dataset. Therefore, we have not wanted to devote much more space to risk management, something that we will address later in a step-by-step process, moving from the most objective and scientific aspects to the more social and management-related aspects. Even so, several mitigation measures are proposed within the scope of the multi-hazard management in Tenerife, in accordance with the scenarios and impacts compiled. Moreover, several entries in the dataset include descriptions of emergency response and recovery, which provide valuable lessons for disaster risk management. We also indicate areas where future research can deepen the integration between empirical event data and risk governance practices.

**Line 386** – *Where is this presented and discussed in the text? Vulnerability is key and inherently difficult to assess, so any insights on vulnerabilities is valuable for DRM.* – The sentence in question referred to the identification of patterns of vulnerability, based on recorded impacts. These include location-specific trends (e.g., recurrent flood damage in low-elevation coastal settlements), severity of impacts (displacement, fatalities, infrastructure damage), and systemic weaknesses in recovery. While we acknowledge that no formal vulnerability index was computed, these descriptive fields offer valuable qualitative insights. We have clarified this in the revised manuscript by explicitly stating that patterns of vulnerability were inferred from observed impacts, not calculated through structured indices.

**Line 388** – *Yes! This is what this paper provides, which is so valuable – possible scenarios, not all scenarios.* – Statement retained and now reinforced with a clearer explanation: this is a scenario-support tool, not a predictive one.

**Line 401** – *hazard not risk.* – Corrected.

**Line 427** – *Is this no. 7 in the list?* – Good point, thank you.

**Line 467** – *Delete – repetition.* – Deleted.

**Line 484** – *It feels a little bit out of place to go into so much detail on citizen science here. Why are you focusing on this in particular? Do you have example where this has really helped? I would prefer to see more focus on community engagement to raise awareness of natural hazard events, potential multi-hazard events that could impact them, and shared learning on how individuals, communities, organisations could increase their own resilience to such events. Also, an important aspect of learning from past events, is local wisdom – learning from people through the stories they share and pass down through generations.* – This is a helpful distinction. Our intention was to refer broadly to citizen participation in the observation, documentation, and interpretation of hazard events—both through formal channels (e.g., intensity questionnaires, landslide reporting apps) and informal, culturally embedded practices (e.g., oral histories, traditional knowledge). To clarify, we have separated "citizen science" (in

the scientific project participation sense) from "community-based engagement", which includes local wisdom, shared learning, and participatory resilience-building. Both are vital, and we now emphasize the complementarity between structured initiatives and community-rooted knowledge systems.

Once again, we thank the reviewer for her insightful and encouraging review. We believe these changes significantly improve the manuscript's clarity, completeness, and utility, and we hope it now meets the high standards expected by the journal.

With kind regards,

**Marta López-Saavedra**
Corresponding author on behalf of all co-authors
August 2025

---

## Author Comment (AC2)

**Author comment to anonymous Referee 1**

Dear Editor and Reviewer 1,

We thank the reviewer for the careful and in-depth analysis of our manuscript. We appreciate the recognition of our work as a valuable empirical contribution and your thoughtful suggestions to strengthen its analytical structure, terminological clarity, and methodological positioning.

We have carefully revised the manuscript in light of your comments. Below, we provide a detailed, point-by-point response.

**General comments**

We appreciate the reviewer's insightful observations, which help clarify the scope and intent of this work. This manuscript should be understood as the very first link in a longer chain of studies that will address the full cycle of multi-hazard data management: compilation, structuring, organization, standardization, processing, analysis, and application. In this initial stage—already representing a substantial effort in terms of research and data gathering—our priority has been to document all hazard-related events exactly as they appear in the historical and bibliographic sources, without reinterpreting or reclassifying them according to existing international frameworks or typologies. This choice was deliberate: applying a classification at this stage would inevitably introduce a degree of subjectivity, which we aim to avoid by designing an automated, data-driven standardization process in later phases.

Our long-term goal is to implement advanced data science techniques—such as machine learning and artificial intelligence—to derive hazard classifications, identify interdependencies, and establish objective, reproducible categories directly from the data itself. While we are fully aware that the current dataset is not yet standardized, interoperable, or ready for direct integration into modelling tools, this step is crucial to ensure that future versions are built on a robust, unbiased foundation. By proceeding in this way, we avoid the risk of producing yet another static database constrained by pre-set typologies, and instead lay the groundwork for a flexible, evidence-based resource that can be adapted to evolving risk governance and technological contexts.

**Section 2: Geological context and natural hazards**

We thank the reviewer for this observation. We respectfully acknowledge the relevance of other eruptions in the Canary Islands and, especially, the 2021 Tajogaite eruption in La Palma as a recent and high-impact event in the region. However, we would like to clarify the rationale behind the structure and scope of Section 2.

This section is intentionally designed to transition from a broad-scale (archipelagic) overview to a focused discussion of Tenerife, the study area of the paper and the sole island covered by the dataset. While we agree that La Palma's eruption is notable, including it without also referencing the 13 other historical eruptions in the Canary Islands would introduce a selective bias. Moreover, assessing whether Tajogaite was the "most damaging" eruption in Canary Islands history is debatable. Such a claim would require a standardized evaluation of direct and

indirect losses (economic, social, environmental), adjusted over time for inflation, population, and development levels—metrics that are not comprehensively available for all historical eruptions. The eruption of Lanzarote in 1730-1736, for example, with almost two cubic kilometers of magma emitted, was much worse and forced the almost total evacuation of the island. For this reason, and to preserve historical consistency and avoid overemphasizing recent events, we have not included it in this section.

According to that, we think it is necessary to focus on our case study, since each island has its own particularities, and although we can give a more general introduction, we don't think it would be necessary to detail other events on the other islands, because then we would also have to talk about floods, landslides, seismicity, not just eruptions. That said, in response to the reviewer's suggestion and to provide additional context, we have added one sentence referencing the regional volcanic hazard across the archipelago and acknowledging that recent eruptions, such as that of Tajogaite (2021), underscore the ongoing relevance of volcanic risk. This maintains the Tenerife-centric scope of the manuscript while acknowledging broader processes.

On the other hand, we agree with the reviewer that the section could better highlight interactions among natural hazards (geological, hydrological, and weather-related hazards), especially those common in Tenerife (e.g., seismicity triggering landslides, eruptions following increased seismic swarms). However, we believe such patterns should emerge from systematic analysis of the dataset, rather than being pre-emptively inferred from anecdotal or visual inspection. As such, we have deliberately avoided subjective interpretation of event relationships unless they are explicitly documented in historical records.

In the revised manuscript, we have added a brief paragraph at the end of the (now) geographical, geological and climatic context section indicating that several types of interactions were captured in the dataset—particularly seismic-volcanic and rainfall-landslide interactions—and that future work will explore these using quantitative methods such as event trees and network analysis.

Regarding the recent seismic uptick, we have chosen not to emphasize this further. The database spans over 500 years, during which numerous similar episodes of seismic unrest have occurred. Highlighting the most recent instance could unintentionally bias the temporal neutrality of the analysis, especially when such episodes have not (yet) resulted in eruptions or documented impacts. Instead, we now reiterate in the conclusion that the dataset is designed to be periodically updated, and future additions will incorporate both newly documented events and new types of analysis, including temporal clustering.

As you suggested, we have renamed and restructured Section 2 as "Geographical, geological and climatic context" of Tenerife, integrating both physical and meteorological elements, to avoid the impression that the study's context is exclusively volcanic or geological. This change improves the internal coherence of the manuscript and addresses the reviewer's concern about focus.

**Section 3. Natural hazards, risk management and risk regulation**

We thank the reviewer for this observation and acknowledge that in the current version, the purpose of introducing the PEIN (2020) risk classification may not be sufficiently clear. The intention was not to replicate PEIN's methodology as part of our analysis, but rather to set the institutional context:

- to show how hazards are formally prioritized in the official risk management framework of Tenerife, and
- to illustrate which hazards are currently of greatest concern to the local government.

We agree that the link to our own work needs to be made explicit. The PEIN (2020) identifies a broad spectrum of natural hazards for Tenerife, including volcanic eruptions, floods, ruptures of storage infrastructures, earthquakes, tsunamis, snowfall, torrential rains, hailstorms, frost, strong winds, coastal storms, heatwaves, haze/dust, droughts, rockfalls, landslides, and coastal erosion.

For the purpose of this study, the dataset focuses on five hazard types:

- Volcanic eruptions
- Earthquakes
- Tsunamis
- Landslides and rockfalls
- Floods

This narrower selection is based on three criteria:

1. Documented historical occurrence (1494–2020) – Only hazards with recurrent, well-documented events over at least several centuries were included, ensuring consistency and reliability in long-term analysis.
2. Availability and quality of historical records – Hazards such as hailstorms, heatwaves, or droughts are relevant but lack sufficiently detailed and consistent historical documentation to support robust analysis in this version of the dataset.
3. Cascading and multi-hazard potential – Selected hazards are known to trigger or be triggered by other events, making them particularly relevant for multi-hazard analysis.
4. These hazards are the ones covered by special plans designed to manage this type of hazard and any resulting emergencies.

This justification has been included in the Methodology section. We emphasize that this is not a judgement on the relative importance of excluded hazards for present-day or future risk management. On the contrary, we recognize that some hazards not included here may gain prominence under climate change scenarios. Our scope is determined by the nature and completeness of available historical data, not by an institutional priority ranking.

Future iterations of the dataset could integrate additional hazards from the PEIN list as historical records are expanded, digitized, or supplemented with palaeoenvironmental or instrumental datasets.

On the other hand, we agree with the reviewer that the analytical flow can be improved by restructuring this section. In the revised manuscript:

1. The climatic context of Tenerife (current weather regimes, precipitation patterns, and hydrometeorological drivers) has been moved from Section 3 to Section 2, which now provides a comprehensive geographical, climatic, and geological context for the island. This allows readers to first understand present-day climatic conditions before considering hazard patterns.

2. The PEIN risk classification remains in Section 3, but the section has been reframed to focus on how hazards are officially recognized, categorized, and prioritized by the Cabildo de Tenerife. We now provide a clearer explanation of the institutional role of the Cabildo as the insular authority responsible for risk management, and how the PEIN integrates into the broader Canary Islands and national DRR governance architecture. We also note that the current planning framework is predominantly single-hazard and does not yet incorporate an integrated multi-hazard approach—reinforcing the relevance of our historical dataset.

This restructuring clarifies the purpose of each component:

- Section 2 – the physical and climatic setting that explains hazard occurrence.
- Section 3 – the institutional hazard inventory and prioritization that forms the starting point for our historical hazard analysis.

Finally, we agree that briefly outlining the broader governance structure can strengthen the contextual foundation of the paper and reinforce the dataset's relevance. While an in-depth governance analysis is beyond the scope of this manuscript, we have now included a short paragraph summarizing how natural hazard risk management is organized in the Canary Islands, the role of the PEIN within this system, and the predominance of single-hazard planning approaches. In addition, reasoning has been added to the justification section that links the information that can be extracted from the PEIN with the results of our analysis of the dataset.

This addition clarifies that our historical multi-hazard dataset provides a foundational evidence base to support future integrated multi-hazard strategies. It also makes explicit why we begin Section 3 with the PEIN hazard list and classification: to anchor our analysis in the current institutional perception and management of risk.

**Section 4. Methodology**

We thank the reviewer for pointing out the inconsistency in hazard classification. This was inherited from the way some special emergency plans in the Canary Islands are presented, where certain hazards are grouped under "geological" plans despite being of hydrological or meteorological origin. Since these plans informed part of our hazard selection, the classification was initially transcribed without modification. The manuscript has now been corrected to distinguish between *geological* hazards (e.g., volcanic eruptions, earthquakes, landslides, rock falls) and *hydrological* hazards (e.g., floods), in line with the PEIN and

international usage. The term "hydrological" has been preferred over "hydrometeorological" to account for other types of triggering events such as tsunamis or dam failures.

Regarding the integration of existing multi-hazard frameworks, we agree that distinctions such as primary/secondary hazards, triggering relationships, and cascading or compound events are essential for multi-hazard risk analysis. However, this dataset was intentionally designed as a first, objective, data-gathering step, recording all identified events in the historical record of Tenerife without interpretive classification of interrelationships. This ensures that the dataset captures the full spectrum of events—regardless of whether their connections are already known or remain to be discovered—avoiding bias in the initial compilation phase. Some cascading events have been labelled as such only when explicitly identified in the source literature, but in most cases, relational patterns will be established through subsequent analysis.

We have revised the text to:

1. Clarify the rationale for this approach;
2. Explicitly situate our methodology as the foundation for later alignment with established hazard typologies and frameworks (e.g., UNDRR, Gill & Malamud 2016, Kappes et al. 2012, López-Saavedra & Martí 2023);
3. Highlight that interoperability with other datasets is a planned next step once analytical processing of these historical records is complete.

**Section 5. Results**

We appreciate the reviewer's comments, and the inconsistencies in the figure numbers have been corrected, as well as some of the references mentioned have been added.

The generation of other figures, such as temporal evolution and spatial distribution, although very useful, are reserved for later processing and analysis, as mentioned above. An approximation has been made in the discussion, but they are outside the scope of the paper, so we have limited ourselves to a preliminary description of the dataset without going into detail, as it is also necessary to first look at cause-effect relationships, patterns, and interrelationships, among other things.

The words "risk" and "hazard" have been reviewed and some corrected, but we confirm that they have been used consciously in accordance with what was intended to be expressed according to their meaning: hazard for the phenomenon, and risk for its potential damage or damaged caused.

Regarding the consequences of landslides and rock falls mentioned in line 320, the reviewer is correct and we agree that a death should entail a cost, if not economic, then certainly a much greater social one, however we have limited ourselves to describing what has been compiled from the written press. In the case of landslides in ravines and on beaches, compared to other events such as eruptions, in many cases there have been no large-scale evacuations or preventive evacuations prior to the danger. Perhaps temporary closure of the area and this is already included in the dataset, which should always be considered as the source of

information, not just the main text. Even so, perhaps we did not see at the time the sensitivity that the reviewer mentions, and for that reason, we wanted to add a clarification in the text so that there are no misunderstandings.

As for the types of interrelationships, as mentioned above, this will be left until the data processing is carried out and the exact types of patterns in the dataset are known. In the case of the sentence indicated by the reviewer on line 317, the sentence has been adapted in the hope that it will now be better understood, but what is meant is that the landslides or rockfalls recorded in the dataset have not triggered other events in a cascade effect, as is clearly recorded in the literature consulted for other hazards.

In the case of the relationship between floods and weather patterns, it is true that this is not mentioned above, but for the same reason we mentioned earlier: to avoid establishing relationships that have not emerged from the observation and processing of the dataset, so as not to reach conclusions based on subjectivity. In section 2, when discussing tropical storms and Atlantic squalls, the type of phenomena they produce, such as unstable weather and precipitation, is already mentioned, and then in the results section, a coincidence of floods with the seasons when such unstable weather occurs is described, hence the relationship.

Regarding the comment on line 332, a sentence has already been added to clarify this reference to "preceding events."

**Section 6. Discussion**

We appreciate the reviewer's suggestion to further connect the recommendations with multi-hazard contexts and to consider potential synergies or trade-offs. While the primary aim of this study was to develop and present the historical multi-hazard dataset, rather than to conduct a full policy analysis, we agree that briefly acknowledging how some measures may interact with other hazards can enhance the discussion. We have therefore added a short paragraph noting these interrelations, and referencing examples from the literature where similar measures have been applied in comparable insular or volcanic regions. This addition preserves the descriptive nature of our work while strengthening the linkage between the dataset findings, the proposed measures, and broader DRR strategies.

Errors in references to figures have been corrected. Additional bibliographic references have also been added to support some statements, but this point has not been expanded upon much further, as these recommendations are based on experience or expert opinion and it has already been made clear in the corrected text that they require analysis of their applicability before being implemented.

Given that the dataset focuses exclusively on volcanic eruptions, earthquakes, floods, landslides/rockfalls, and tsunamis, we cannot expand further on other hazards because, on the one hand, we do not have and have not collected sufficient data to analyze them objectively and treat them appropriately without relying on our own experience and knowledge of the region and, on the other hand, because it exceeds the scope of the article and would make it too long. For this reason, they have been mentioned briefly so as not to overlook them, and its

potential to amplify or cause other hazards has already been highlighted in the text, but they couldn't more thoroughly analyzed

On the other hand, we acknowledge the reviewer's comment and agree that clarifying the current limitations of the dataset will strengthen the manuscript. While the dataset is already structured to allow for future applications in risk modelling and machine learning, its present form should be understood as a first version, focused on compilation and harmonisation of historical data in order to standardize it in next steps. Explicitly recognising the current gaps in metadata standardization, indicators, and validation protocols will help set realistic expectations and guide future improvements, for that reason we have included a paragraph at the end of the conclusion section.

As for vulnerability and socioeconomic dynamics, this will come much later. This document should be understood as the first piece in a whole chain that will cover the entire process, so addressing it here without being rigorous with the methodology could lead to results or treatment of this information that, as is often the case, would be subject to subjectivity or would simply remain as recommendations, something we do not want. Nevertheless, we appreciate this contribution, as we consider it very valuable and it reinforces our intention to continue with this work.

Once again, we thank the reviewer for her insightful and encouraging review. We believe these changes significantly improve the manuscript's clarity, completeness, and utility, and we hope it now meets the high standards expected by the journal.

With kind regards,

**Marta López-Saavedra**
Corresponding author on behalf of all co-authors
August 2025